# Disentangling Length Bias in Preference Learning via Response-Conditioned Modeling

**Jianfeng Cai**[†]                **Jinhua Zhu**[†]                **Ruopei Sun**[†]

**Yue Wang**[‡]        **Li Li**[†]        **Wengang Zhou**[† *]        **Houqiang Li**[† *]

[†]University of Science and Technology of China    [‡]Zhongguancun Academy
{xiaobaicai,teslazhu}@mail.ustc.edu.cn

## Abstract

Reinforcement Learning from Human Feedback (RLHF) has achieved considerable success in aligning large language models (LLMs) by modeling human preferences with a learnable reward model and employing a reinforcement learning algorithm to maximize the reward model's scores. However, these reward models are susceptible to exploitation through various superficial confounding factors, with length bias emerging as a particularly significant concern. Moreover, while the pronounced impact of length bias on preference modeling suggests that LLMs possess an inherent sensitivity to length perception, our preliminary investigations reveal that fine-tuned LLMs consistently struggle to adhere to explicit length instructions. To address these two limitations, we propose a novel framework wherein the reward model explicitly differentiates between human semantic preferences and response length requirements. Specifically, we introduce a **R**esponse-**c**onditioned **B**radley-**T**erry (Rc-BT) model that enhances the model's capability in length bias mitigating and length instruction following, through training on our augmented dataset. Furthermore, we propose the Rc-RM and Rc-DPO algorithm to leverage the Rc-BT model for reward modeling and direct policy optimization (DPO) of LLMs, simultaneously mitigating length bias and promoting adherence to length instructions. A series of experiments across various models and datasets demonstrate the effectiveness and generalizability of our approach.

## 1 Introduction

Reinforcement learning from human feedback (RLHF) (Christiano et al., 2017; Ziegler et al., 2019; Ouyang et al., 2022) has revolutionized the field of natural language processing, enabling advancements in areas such as conversation (Bai et al., 2022a), code generation (Poesia et al., 2022; Jiang et al., 2024), complex planning (Hao et al., 2023; Zhao et al., 2024), mathematical reasoning (Imani et al., 2023; Luo et al., 2023), and so on. Within this framework, preference learning plays a pivotal role by using a generalizable model as a proxy for human preferences and optimizing large language models (LLMs) based on this preference model. However, a significant challenge in RLHF is the phenomenon termed reward overoptimization or hacking (Gao et al., 2023). This occurs when excessive optimization against the preference model undermines the attainment of the true objective. As a result, LLMs may learn to exploit simpler criteria such as length, bullet points, or politeness (Ramé et al., 2024), rather than more causal and nuanced indicators to achieve higher reward.

Among these reward hacking phenomena, length bias stands out as both a prevalent and challenging issues (Dubois et al., 2024a), and we identify it as a representative test bed, with the expectation that our approach can be readily extended to address other spurious correlations. To mitigate length bias in the RLHF pipeline, two primary strategies have emerged, which may be complementary: The first strategy focuses on rectifying the policy optimization process through techniques such as increasing the KL penalty coefficient, applying length penalty rewards (Singhal et al., 2023),

---

[*]Corresponding authors.

reward clipping (Chen et al., 2024b), and various hyper-parameter tuning methods. The second strategy aims to disentangle length information from quality in reward modeling. This includes length balancing, reward data augmentation, confidence-based truncation in training data (Singhal et al., 2023), and using sophisticated objectives to separate length information (Chen et al., 2024b).

Despite these advances, we observe several drawbacks in these methods. First, rectification methods for policy optimization are sensitive to hyperparameter changes and foundational model variations, and prior work demonstrates that their effectiveness is limited (Singhal et al., 2023; Chen et al., 2024b). The second category shows superior performance compared to the first; however, over-parameterization of the two branches can cause unstable optimization (Kurach et al., 2019), and regularization methods that encourage linear irrelevance may not guarantee true independence (Shimony, 1993). Furthermore, ignoring the biased nature of the training data and forcing an irrelevancy or orthogonality between response quality and response length is not always reasonable. For instance, in some instruction-following datasets, such as length instruction dataset, response length may indeed be a factor of response quality. Finally, these methods regard length information as harmful and aim to minimize its influence on the instruction-following. However, whether we could leverage these length biases for better preference modeling remains an underexplored question.

Based on the aforementioned observations, we aim to improve preference modeling by leveraging response length information and propose a novel approach that allows the preference model to explicitly differentiate between human semantic intent and response length instructions. Specifically, we first develop an augmented length instruction dataset derived from the original preference dataset. Subsequently, we introduce a Response-conditioned Bradley-Terry (Rc-BT) model, which improves the model's ability to mitigate length bias and follow length instructions.

Given the widespread use of paired data preference modeling in RLHF, our method can be seamlessly integrated into both the reward modeling and policy optimization stages. First, we empirically verify our method in the reward modeling task. The results demonstrate that our method not only enhances the reward model's ability to mitigate length bias, thereby improving its alignment with human semantic quality, but also strengthens its adherence to length instructions. Furthermore, applying our method to direct preference optimization (DPO) (Rafailov et al., 2024), a popular alignment algorithm, further validates its effectiveness.

Our contributions are summarized as follows: (1) We propose the Response-conditioned Bradley-Terry method to mitigate length bias and enhance the model's capacity to follow length instructions. (2) We show that our method can be integrated into reward modeling and policy optimization with minimal adjustments. (3) Experimental results illustrate the superiority and generalizability of our method across multiple models and datasets.

## 2 RELATED WORK

In this section, we briefly introduce the background of RLHF, reward hacking and length instruction following. A more detailed version is in Appendix B.

**Reinforcement Learning From Human Feedback.** RLHF (Christiano et al., 2017) involves training a reward model to approximate human preferences and optimizing the LLMs through reinforcement learning (RL) (Schulman et al., 2017). However, this approach suffers from training instability and requires careful tuning of numerous hyperparameters. Recent direct preference alignment methods, particularly DPO (Rafailov et al., 2024), offer a more stable alternative. By reformulating the reward function, DPO eliminates the need for an online reward model, enabling robust offline preference learning (Hong et al., 2024; Chen et al., 2024a; Ethayarajh et al., 2024). Our method not only enhances RLHF by improving reward model but also seamlessly integrates into the DPO framework.

**Reward Hacking.** RLHF is vulnerable to reward hacking, where LLMs exploit inherent biases in the proxy preference model to achieve higher scores (Pan et al., 2022; Casper et al., 2023; Lambert & Calandra, 2023). This phenomenon persists in DPO, primarily arising from task complexity, evaluation limitations, and biased feedback (Dubois et al., 2024b). One common form of reward hacking is length bias (Singhal et al., 2023; Park et al., 2024), wherein models favor longer responses regardless of semantic quality. Previous attempts to address this issue include ODIN's (Chen et al., 2024b) length regularization in reward modeling; dual-model training with different learning rates (Shen et al., 2023); and DPO objective modification with length penalties (Park et al., 2024). In contrast to

these methods that suppress length information, our method enables models to distinguish between semantic intent and length instructions, preserving both aspects in a balanced manner.

**Length Instruction Following.** Current LLMs can respond to qualitative length descriptors like "concise" or "verbose" but struggle with explicit numerical constraints such as "150 words or less" (Yuan et al., 2024). Although LIFT (Yuan et al., 2024) improves length instruction adherence through specialized datasets, this approach compromises semantic quality, resulting in degraded overall performance. Our method, in contrast, achieves effective length instruction following while simultaneously enhancing semantic quality.

## 3 PRELIMINARY EXPLORATIONS

In this section, we conduct several preliminary investigations into the ability of LLMs to perceive and process length information, with an emphasis on the reward model. Following Yuan et al. (2024), we utilize the OpenAssistant dataset (Köpf et al., 2024), which is partitioned into three subsets: $\mathcal{D}_{sft}$ for supervised fine-tuning (SFT), $\mathcal{D}_{rm}$ for reward model (RM) training, and $\mathcal{D}_{eval}$ for RM evaluation. Additionally, we employ a range of models for subsequent experimental analyses, including Qwen2-1.5B, Qwen2.5-7B, and Llama-3.1-8B. Additional details of the training and evaluation procedures are provided in Appendix D and summarize all notations in Appendix A to aid comprehension.

### 3.1 LENGTH BIAS INDEED EXISTS

To verify the presence of length bias in reward models trained on $\mathcal{D}_{rm} = \{(x^{(i)}, y_c^{(i)}, y_r^{(i)})\}$, where $x^{(i)}$ is the prompt and response $y_c^{(i)}$ is preferred to $y_r^{(i)}$ according to human annotations, we design three additional evaluation datasets based on $\mathcal{D}_{eval}$: Each prompt $x^{(i)}$ within $\mathcal{D}_{eval}$ is replaced with an empty prompt $x_e =$ *"empty prompt"*, forming the empty evaluation dataset $\mathcal{D}_{eval}^e$; Each prompt $x^{(i)}$ is randomly replaced with $x^{(j)}$ (where $i \neq j$)[1], resulting in the random evaluation dataset $\mathcal{D}_{eval}^r$; building upon the dataset $\mathcal{D}_{eval}^r$, randomly shuffling the token sequences of each response $y_c/y_r$ in the preference pairs to create a random sequence evaluation dataset $\mathcal{D}_{eval}^{s,r}$.

Table 1: Evaluation results of reward models on different evaluation datasets $\mathcal{D}_{eval}$, $\mathcal{D}_{eval}^e$ and $\mathcal{D}_{eval}^r$. (Accuracy: percentage of preference pairs correctly ranked by the reward model. Consistency: percentage of preference pairs ranked identically by the reward model across different datasets.)

| Model | Accuracy (%) | | | | Consistency (%) |
|---|---|---|---|---|---|
| | $\mathcal{D}_{eval}$ | $\mathcal{D}_{eval}^e$ | $\mathcal{D}_{eval}^r$ | $\mathcal{D}_{eval}^{s,r}$ | $\mathcal{D}_{eval}^e$ ($\mathcal{D}_{eval}^r$) |
| Qwen2-1.5B-Base | 63.86 | 64.13 | 64.40 | 58.14 | 89.40 (87.34) |
| Qwen2-1.5B-Instruct | 62.77 | 65.22 | 60.87 | 57.88 | 92.12 (90.42) |
| Qwen2.5-7B-Base | 57.07 | 60.05 | 60.33 | 56.97 | 88.32 (90.13) |
| Qwen2.5-7B-Instruct | 63.04 | 62.23 | 60.60 | 57.12 | 88.60 (89.85) |
| Llama-3.1-8B-Base | 56.25 | 60.05 | 61.41 | 56.43 | 89.40 (85.24) |
| Llama-3.1-8B-Instruct | 57.88 | 58.15 | 56.25 | 57.07 | 88.59 (87.26) |

Table 1 reports the evaluation results. Despite the semantic mismatch between responses and prompts in $\mathcal{D}_{eval}^e$, $\mathcal{D}_{eval}^r$, and $\mathcal{D}_{eval}^{s,r}$, the reward model nevertheless attains relatively high accuracy on these datasets. Specifically, nearly all models achieve an accuracy exceeding $60\%$, which is very close to the accuracy observed on the original dataset $\mathcal{D}_{eval}$. This finding suggests that there is a significant bias in the models towards preferring $y_c$, even when both responses $y_c$ and $y_r$ are misaligned with the prompt.

Next, we examine the consistency of evaluation results across different reward models between $\mathcal{D}_{eval}^e$ (or $\mathcal{D}_{eval}^r$) and $\mathcal{D}_{eval}$, with a particular focus on the influence of varying prompts on the model's preferences. The results, as shown in Table 1, reveal that the models exhibit the same preference for responses in over $85\%$ of cases, despite different prompts. This suggests that the models' preferences are not primarily driven by the prompts.

---

[1] For clarity of notation, we omit the superscript $^{(i)}$ in subsequent discussions where no ambiguity exists.

We further plot the relationship between response length and reward score for the reward model across $\mathcal{D}_{eval}$, $\mathcal{D}_{eval}^e$, $\mathcal{D}_{eval}^r$, and $\mathcal{D}_{eval}^{s,r}$, as shown in Figure 1. A strong positive relationship is observed, suggesting that the reward model heavily relies on response length as a criterion for evaluating response quality. These findings provide compelling evidence of length bias in the reward models. More detailed analysis can be found in Appendix E.1.

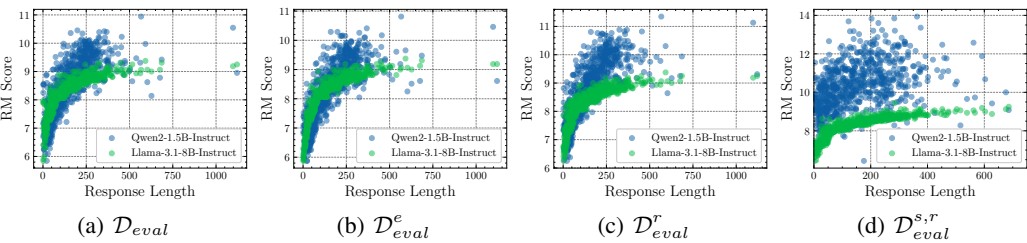

(a) $\mathcal{D}_{eval}$  (b) $\mathcal{D}_{eval}^e$  (c) $\mathcal{D}_{eval}^r$  (d) $\mathcal{D}_{eval}^{s,r}$

Figure 1: The relationships between response lengths and scores of reward models (Baseline) trained with Qwen2-1.5B-Instruct and Llama-3.1-8B-Instruct, evaluated on different evaluation datasets.

## 3.2 LENGTH BIAS IN EVALUATION DATASET

Here, we aim to verify whether the initial evaluation dataset $\mathcal{D}_{eval}$ exhibits length bias, which could hinder its ability to accurately evaluate the true performance of the reward models. Specifically, our analysis reveals that 59.78% of the chosen responses in $\mathcal{D}_{eval}$ are longer than the corresponding rejected responses. Consequently, this length bias allows the model to attain an accuracy nearing 60% merely by favoring longer responses, resulting in an unreliable evaluation of the model.

To fairly and accurately assess the semantic quality of the reward model, we reconstruct the quality evaluation dataset $\mathcal{D}_{eval}^q$ from $\mathcal{D}_{eval}$ through an automated process. Specifically, we employ GPT-4o (Hurst et al., 2024) to transform each triplet $(x, y_c, y_r)$ in $\mathcal{D}_{eval}$ into two distinct triplets $(x, y_c^1, y_r^1)$ and $(x, y_c^2, y_r^2)$. These newly generated triplets adhere to the length constraints $|y_r^1| > |y_c^1|^2$ and $|y_r^1| < |y_c^2|$, while ensuring that $y_c^1$ and $y_c^2$, as well as $y_r^1$ and $y_r^2$, remain semantically similar to the original $y_c$ and $y_r$. The detailed generation process is in Appendix D.2.

The reconstructed evaluation dataset $\mathcal{D}_{eval}^q$ demonstrates significant improvement in mitigating the inherent length bias in $\mathcal{D}_{eval}$, thereby facilitating a more objective comparison among diverse reward models. As evidenced in Table 4, under this less biased evaluation dataset, Baseline models exhibit notably reduced performance metrics, with the majority of accuracy scores not exceeding 60%. Further experimental analysis of $\mathcal{D}_{eval}^q$ is detailed in Appendix E.2.

## 3.3 LENGTH BIASED MODEL SHOW LIMITED ADHERENCE TO LENGTH INSTRUCTIONS

The reward model demonstrates length bias, indicating its ability to perceive response length. However, the question remains whether it can apply this ability to follow length instruction $x_l$, where $x_l = length\ constraint + x$. For simplicity, we use the same *length constraint* as LIFT (Yuan et al., 2024), which specified by *word_num*, as detailed in Appendix D.4. To validate this, we construct the length evaluation dataset $\mathcal{D}_{eval}^l$ based on $\mathcal{D}_{eval}^q$, where both responses semantically satisfy $x$ in $x_l$, but only one adheres to the length constraint. Appendix D.3 for more construction details.

When evaluated on $\mathcal{D}_{eval}^l$ (results shown in Table 10, Baseline), all models demonstrate the accuracy close to 50%, no better than chance. This demonstrates that models with length bias are unable to apply this ability to follow length instructions.

We further implemented a simple heuristic evaluation to serve as an upper bound. Specifically, for each triplet $(x^{(i)}, y_c^{(i)}, y_r^{(i)})$, we replaced the original prompt $x^{(i)}$ with a length-only instruction ("Please generate a response more/less than *word_num* words") and ensured that $y_c^{(i)}$ and $y_r^{(i)}$ were approximately balanced in terms of satisfying or violating the constraint, generating heuristic evaluation datasets $\mathcal{D}_{eval}^{h,l}$. We then evaluated the Baseline models, based on Qwen2.5-7B-Instruct and

---

$^2|y|$ denotes the length of a given response $y$.

Llama3.1-8B-Instruct, on this dataset to assess their compliance with length instructions. The results as shown in Table 2. The results reveal that even under such simplified conditions, the Baseline models exhibit negligible ability to follow explicit length instructions.

Table 2: Evaluation results of reward models on heuristic evaluation datasets $\mathcal{D}_{eval}^{h,l}$.

| Metrics | Qwen2.5-7B-Instruct | Llama-3.1-8B-Instruct |
|---|---|---|
| heuristic (%) | 56.73 | 54.12 |

This finding suggests that reward models unconsciously acquire length bias during preference learning without explicit awareness of length as a measurable attribute. Motivated by this observation, we propose the following hypothesis:

**Hypothesis 1.** *Explicit length-instruction learning enables language models to form a clearer perception of target response length, thereby reducing undesired length bias.*

Based on this, we propose incorporating length constraints into the training process to transform implicit length bias into explicit length understanding.

### 3.4 LENGTH INSTRUCTIONS ARE EASILY LEARNED

The most straightforward approach to explicit length instruction learning would be training directly on data formatted as $\{x_l, y_c, y_r\}$. To examine the effectiveness of this intuitive format, we extend LIFT (Yuan et al., 2024) to create LIFT-plus by incorporating minimum length constraints ("or more" length instruction) alongside its original maximum length constraints ("or less" length instruction)[3]. We then construct three variant datasets: LIFT-plus$_2^{reverse}$, which reverses the preference order between chosen and rejected responses based on length constraints; LIFT-plus$_2^{noreverse}$, which preserves the original preference ordering; and LIFT-plus$_2^{empty}$, where substitutes $x$ in $x_l$ with an empty prompt $x_e$ while maintaining the preference order, examines the impact of semantic prompts on length instruction following. The complete construction process is in Appendix D.4.

Empirical results presented in Table 9 (Appendix E.3) reveal that while models demonstrate notably higher accuracy on $\mathcal{D}_{eval}^l$, they achieve diminished accuracy on $\mathcal{D}_{eval}^q$ compared to the Baseline in Table 4. This performance disparity suggests that reward models prioritize length instruction compliance at the expense of semantic understanding. Furthermore, accuracy trajectories across training steps (Figure 5, Appendix E.3) indicates a similar pattern: accuracy on $\mathcal{D}_{eval}^l$ exhibits rapid improvement, while accuracy on $\mathcal{D}_{eval}^q$ shows initial improvement followed by deterioration.

These results highlight the limitations of the $\{x_l, y_c, y_r\}$ format, which encourages overfitting to length instructions at the expense of semantic capability, indicating the need for better methods.

## 4 RESPONSE-CONDITIONED MODELING

In this section, we present our Response-conditioned Bradley-Terry model, with the overall framework depicted in Figure 2. In Section 4.1, we briefly review the Bradley-Terry (BT) model. In Section 4.2, we introduce the **R**esponse-**c**onditioned **B**radley-**T**erry (Rc-BT) model[4], presenting its algorithmic details and mathematical formulation. Finally, we demonstrate the implementation in both Reward Model (Section 4.3) and DPO (Section 4.4) through mathematical derivations.

### 4.1 PRELIMINARY: BRADLEY-TERRY MODEL

In RLHF, given a human request $x$, a supervised fine-tuned LLM denoted as $\pi^{\text{SFT}}$ is prompted with $x$ to sample response pairs $y_1$, $y_2$. Human feedback on this pair is then collected and represented as $y_c \succ y_r$, where $y_c$ is preferred over $y_r$. To model human response preferences, the Bradley-Terry

---

[3]During the improvement over LIFT, the training dataset always includes the original dataset $\mathcal{D}_{rm}$ to ensure a balanced exposure to both original and length-instruction formatted data.

[4]While primarily developed for length bias mitigation and length instruction adherence, the Rc-BT framework is theoretically applicable to diverse instruction-following tasks.

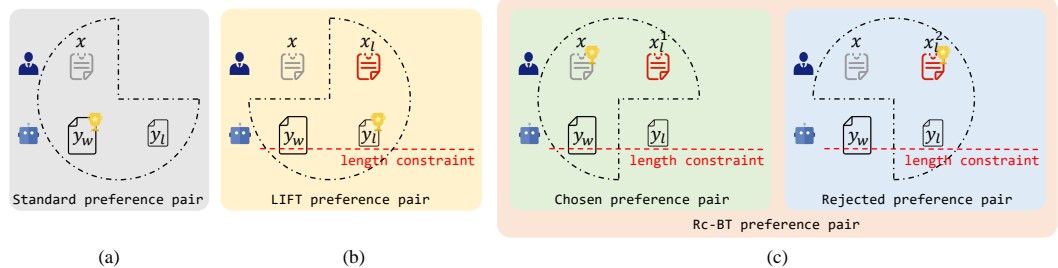

Figure 2: To illustrate the distinct data formats across different methods, we present the maximum length instruction case: **(a)** The conventional RLHF (Ouyang et al., 2022) with standard preference pair $(x, y_c, y_r)$ ; **(b)** LIFT with augmented format $(x_l, y_r, y_c)$; and **(c)** Our method (Rc-BT) with two preference pairs $(x, x_l^1, y_c)$ and $(x_l^2, x, y_r)$. The term length constraint refers to the maximum allowable length of the response as specified in the length instruction $x_l$, $x_l^1$, or $x_l^2$. The black dashed lines indicate the data utilized by each method.

(BT) (Bradley & Terry, 1952) model is a prevalent choice (Ouyang et al., 2022). It models the underlining human preference distribution $p^*$ by assuming an underlying true reward model $r^*$:

$$p^*(y_c \succ y_r) = \frac{\exp\left(r^*(x, y_c)\right)}{\exp(r^*(x, y_c)) + \exp(r^*(x, y_r))}, \tag{1}$$

where $r^*$ is not accessible or easily defined by several rules. Therefore, people try to parameterize a reward model $r_\phi$ and optimize the parameters via maximum likelihood or leverage an analytical mapping from reward function to optimal policy, to directly optimize LLM policy (Dubey et al., 2024). Given a high quality preference dataset $\{(x, y_w, y_r)\}$, the BT model has demonstrated significant efficacy in various preference modeling and policy optimization tasks (Ji et al., 2023).

## 4.2 RESPONSE-CONDITIONED BRADLEY-TERRY MODEL

Response preference modeling employs Eqn. 1 as a framework for response differentiation. However, a fundamental challenge arises in scenarios involving prompts with multiple instructions, where responses may only partially satisfy the given criteria. As previously discussed in Section 3.4, the conventional $\{x_l, y_c, y_r\}$ data format exhibits a tendency to overfit to length instructions, consequently compromising the model's semantic comprehension capabilities. This limitation represents an inherent constraint in response preference modeling.

Therefore, we propose a *response-conditioned modeling* framework, as illustrated in Figure 2. Our approach proceeds as follows: given an original chosen response $y_c$ and prompt $x$, we construct a length augmented instruction $x_l^1$ by incorporating a *length constraint* with the prompt $x$, ensuring that $y_c$ violates this constraint. We then formulate a preference pair $(x, x_l^1, y_c)$, where $(x, y_c)$ is considered preferable to $(x_l^1, y_c)$. This preference is then modeled using the BT model as follow:

$$p^*(x \succ x_l^1 | y_c) = \frac{\exp\left(r^*(x, y_c)\right)}{\exp(r^*(x, y_c)) + \exp(r^*(x_l^1, y_c))}. \tag{2}$$

Similarly, for a given rejected response $y_r$ and original prompt $x$, we construct a length augmented instruction $x_l^2$ such that $y_r$ satisfies the specified length constraint. We then formulate a preference pair $(x_l^2, x, y_r)$, where $(x_l^2, y_r)$ is considered preferable to $(x, y_r)$. This preference structure is similarly modeled using the BT formulation:

$$p^*(x_l^2 \succ x | y_r) = \frac{\exp\left(r^*(x_l^2, y_r)\right)}{\exp(r^*(x_l^2, y_r)) + \exp(r^*(x, y_r))}. \tag{3}$$

The final response-conditioned modeling preference dataset is defined as $\mathcal{D}_{Rc} = \{(x, x_l^1, y_c)\} \cup \{(x_l^2, x, y_r)\}$. Through maximum likelihood estimation, we derive the Response-conditioned BT (Rc-BT) modeling objective function as follows:

$$\mathcal{L}_{Rc} = - \mathop{\mathbb{E}}_{(x,x_l^1,y_c)\sim\mathcal{D}_{Rc}} [\log p^*(x \succ x_l^1|y_c)] - \mathop{\mathbb{E}}_{(x_l^2,x,y_r)\sim\mathcal{D}_{Rc}} [\log p^*(x_l^2 \succ x|y_r)]. \tag{4}$$

The Rc-BT facilitates explicit comparisons between $x$ and its length-augmented variants ($x_l^1$ and $x_l^2$), enabling the model to systematically perceive response lengths and thus mitigating implicit length bias through explicit length understanding. To better contrast the BT model with our approach, Table 3 provides a detailed comparison between BT and our method Rc-BT.

Table 3: Comparison of Preference Modeling Methods (BT vs Rc-BT).

| Method | BT | Rc-BT |
|---|---|---|
| Motivation | Capture preferences between responses | Capture preferences between prompts |
| Data format | $(x, y_c, y_r)$ | $(x_c, x_r, y)$ |
| Modeling equation | $p^*(y_c \succ y_r) = \frac{\exp(r^*(x,y_c))}{\exp(r^*(x,y_c))+\exp(r^*(x,y_r))}$ | $p^*(x_c \succ x_r) = \frac{\exp(r^*(x_c,y))}{\exp(r^*(x_c,y))+\exp(r^*(x_r,y))}$ |

### 4.3 RESPONSE-CONDITIONED REWARD MODEL

In response preference modeling, the reward model is initialized from $\pi^{\text{SFT}}$ and augmented with a linear projection layer that transforms the complete sequence representation into a scalar value $r_\phi(x, y)$. Given a preference dataset $\mathcal{D} = \{(x, y_w, y_r)\}$, the reward model is optimized through maximum likelihood estimation according to Eqn. 1:

$$\mathcal{L}_{r_\phi}(\mathcal{D}) = - \mathop{\mathbb{E}}_{(x,y_w,y_r)\sim\mathcal{D}} [\log \sigma(r_\phi(x, y_w) - r_\phi(x, y_r))], \tag{5}$$

where $\sigma(\cdot)$ is the sigmoid function. In response-conditioned modeling, the architectural structure of the parameterized reward model $r_\phi$ remains unchanged while the data format is transformed from prompt-conditioned to response-conditioned. Similar to Eqn. 5, the model is optimized through maximum likelihood estimation based on Eqn. 4 using dataset $\mathcal{D}_{Rc}$:

$$\mathcal{L}_{r_\phi}(\mathcal{D}_{Rc}) = - \mathop{\mathbb{E}}_{(x,x_l^1,y_c)\sim\mathcal{D}_{Rc}} [\log \sigma(r_\phi(x, y_c) - r_\phi(x_l^1, y_c))] - \lambda \mathop{\mathbb{E}}_{(x_l^2,x,y_r)\sim\mathcal{D}_{Rc}} [\log \sigma(r_\phi(x_l^2, y_r) - r_\phi(x, y_r))], \tag{6}$$

where $\lambda$ is used to balance the relative contribution of the pairs $(x, x_l^1, y_c)$ and $(x_l^2, x, y_r)$.

### 4.4 RESPONSE-CONDITIONED DIRECT PREFERENCE OPTIMIZATION

In response preference modeling, DPO derives an alternative formulation from Eqn. 9, where the reward is expressed as a function of the optimal policy. By substituting this formulation into the reward optimization objective specified in Eqn. 5, we obtain a direct optimization approach for policy training. Specifically, the policy can be optimized on dataset $\mathcal{D}$ using the following objective[5]:

$$\mathcal{L}_{DPO}(\pi_\theta; \pi_{\text{ref}}) = - \mathop{\mathbb{E}}_{(x,y_c,y_r)\sim\mathcal{D}} [\log \sigma(\beta \log \frac{\pi_\theta(y_c|x)}{\pi_{\text{ref}}(y_c|x)} - \beta \log \frac{\pi_\theta(y_r|x)}{\pi_{\text{ref}}(y_r|x)})]. \tag{7}$$

For the response-conditioned modeling, we follow an analogous derivation process to DPO. Specifically, we begin with the modified RL objective (Eqn. 10), derive the reward model expression in terms of the optimal policy, and subsequently incorporate it into the Rc-BT modeling function (Eqn. 4). This derivation yields the Response-conditioned DPO (Rc-DPO) objective function:

$$\begin{aligned}
\mathcal{L}_{DPO}^{Rc}(\pi_\theta; \pi_{\text{ref}}) = - &\mathop{\mathbb{E}}_{(x,x_l^1,y_c)\sim\mathcal{D}_{Rc}} [\log \sigma(\beta \log \frac{\pi_\theta(x, y_c)}{\pi_{\text{ref}}(x, y_c)} - \beta \log \frac{\pi_\theta(x_l^1, y_c)}{\pi_{\text{ref}}((x_l^1, y_c)})] \\
- &\mathop{\mathbb{E}}_{((x_l^2,x,y_r)\sim\mathcal{D}_{Rc}} [\log \sigma(\beta \log \frac{\pi_\theta((x_l^2, y_r)}{\pi_{\text{ref}}(x_l^2, y_r)}) - \beta \log \frac{\pi_\theta(x, y_r)}{\pi_{\text{ref}}(x, y_r)}].
\end{aligned} \tag{8}$$

For a detailed and complete derivation, please refer to Appendix C.

---

[5]For the remainder of this paper, unless explicitly stated otherwise, every instance of $\pi_\theta$ denotes $\pi_\theta(y \mid x)$.

## 5 EXPERIMENTS

This section presents experiments and ablation studies to demonstrate the effectiveness and validate the design of our Rc-BT framework. We primarily demonstrate the results of our method in mitigating length bias, while its efficacy in length instruction following can be found in Appendix F.

### 5.1 EXPERIMENTAL SETTINGS

**Dataset and Models.** Consistent with Section 3, for the dataset, we use $\mathcal{D}_{sft}$ for SFT to finetune the *Base* models. For RM and DPO, Rc-BT generates an augmented dataset $\mathcal{D}_{Rc}$ derived from $\mathcal{D}_{rm}$. Both reward models and DPO models are trained on the combined dataset $\mathcal{D}_{rm} \cup \mathcal{D}_{Rc}$, referred to as Rc-RM and Rc-DPO, respectively. We employ three pretrained models as our base models: Qwen2-1.5B[6], Qwen2.5-7B, and Llama-3.1-8B, Gemma-2-9B and Qwen2.5-14B.

**Training Details.** For Rc-RM training, the learning rate is set to $1 \times 10^{-5}$, followed by a cosine learning rate schedule with an initial warmup of 10 steps and a batch size of 64. Each experiment is trained for 5 epochs. For Rc-DPO training, the settings are identical to RM training, except the learning rate is $1 \times 10^{-6}$. All experiments are implemented based on DeepSpeed (Yao et al., 2023) and Transformers (Wolf et al., 2020), and conducted on a machine with 8 NVIDIA A100 GPUs.

**Compared Methods and Evaluation.** For RM evaluation, we assess the performance across methods using two primary metrics: *Quality Eval Acc* and *Length Eval Acc*, which are accuracy on $\mathcal{D}_{eval}^{q}$ and $\mathcal{D}_{eval}^{l}$ respectively. The former metric evaluates semantic quality with a focus on length bias mitigation, while the latter measures the effectiveness of length instruction adherence.

For DPO evaluation, leveraging recent advancements in automated assessment approaches (Zheng et al., 2023; Dubois et al., 2024b), we employ a model-based evaluation framework to systematically assess the quality of generated responses. First, we construct the AlpacaEval-LI-plus-less and AlpacaEval-LI-plus-more benchmarks, enhanced versions of AlpacaEval (Dubois et al., 2024b), designed to evaluate the model's ability to follow "or less" and "or more" length instructions, respectively. Then, we evaluate each DPO model's performance by comparing it with its corresponding SFT/*Instruct* model using two complementary metrics. The first metric, *Length Acc*, quantifies the model's adherence to length instructions by measuring how well the generated responses conform to the specified length constraints in $x_l$. The second metric assesses the model's semantic instruction following capability through two comparative win ratios: *Length Win Ratio* represents the proportion of cases where the model generates semantically superior responses when prompted with length-augmented instructions ($x_l$), while *Quality Win Ratio* measures the proportion of cases where the model demonstrates better semantic quality with original prompts ($x$). For comprehensive evaluation details, please refer to Appendix F.1.

We refer to the model trained on the $\mathcal{D}_{rm}$ as **Baseline**. For RM, we incorporate **ODIN** (Chen et al., 2024b) and **R-DA** (Singhal et al., 2023) as comparison method, with particular emphasis on its *Quality Eval Acc*, owing to its demonstrated effectiveness in mitigating length bias. In addition, we compare our method with **LIFT-plus** on the *Length Eval Acc*[7]. For DPO, we include **LIFT-plus**, **R-DPO** (Park et al., 2024), **Dr. DPO** (Wu et al., 2024), **SimPO** (Meng et al., 2024), and **CPO** (Xu et al., 2024a) as comparison methods. Unless otherwise specified, all compared methods are trained using the recommended settings from their papers.

### 5.2 THE RESULTS OF REWARD MODELS

**Rc-RM significantly alleviates length bias.** The *Quality Eval Acc* results in Table 4 clearly demonstrate the effectiveness of Rc-RM in mitigating length bias. Specifically, Rc-RM outperforms Baseline, R-DA, and ODIN across all models. Notably, on Qwen2-1.5B-Base, Rc-RM exceeds Baseline by 10.41% and outperforms ODIN by 13.43%, respectively, on Llama-3.1-8B-Instruct, it surpasses Baseline by 16.85% and ODIN by 11.54%, and on Gemma-2-9B-it, Rc-RM surpasses Baseline by 10.11% and ODIN by 7.71%. Additionally, Qwen2.5-14B-Instruct also benefits significantly from Rc-RM. It achieves a remarkable 81.70% *Quality Eval Acc*, substantially higher than ODIN, R-DA,

---

[6]We also conduct experiments using the Qwen2.5-1.5B, which shown in Appendix F.3

[7]As discussed in Section 3, LIFT-plus harms semantic quality and does not mitigate length bias, so we exclude it from the *Quality Eval Acc* comparison.

Table 4: *Quality Eval Acc* of different reward models on quality ($\mathcal{D}_{eval}^q$) evaluation datasets.

| Model | Baseline | ODIN | R-DA | Rc-RM |
|---|---|---|---|---|
| Qwen2-1.5B-Base | 59.14 | 56.12 | 60.17 | **69.55** |
| Qwen2-1.5B-Instruct | 60.75 | 63.56 | 61.27 | **71.47** |
| Qwen2.5-7B-Base | 54.26 | 60.90 | 63.46 | **70.74** |
| Qwen2.5-7B-Instruct | 59.31 | 67.55 | 66.34 | **73.07** |
| Llama-3.1-8B-Base | 59.04 | 60.37 | 65.17 | **70.51** |
| Llama-3.1-8B-Instruct | 55.59 | 60.90 | 60.78 | **72.44** |
| Gemma-2-9B-it | 53.45 | 55.85 | 55.21 | **63.56** |
| Qwen2.5-14B-Instruct | 65.57 | 76.22 | 75.14 | **81.70** |

and the Baseline, validating Rc-RM's robustness under model scaling. Overall, these results affirm the generalizability and scalability of our Rc-RM framework across diverse model families and sizes, reinforcing its effectiveness in mitigating length-bias.

We conducted further experiments by training a policy using the trained reward models via PPO. We trained the *Qwen2-1.5B-Instruct* policy using the reward models of Qwen2-1.5B-Instruct and Qwen2.5-7B-Instruct from Table 4. The evaluation was carried out on AlpacaEval following the same metrics used for evaluating the DPO models. The results are in Table 11 (Appendix F.2). It can be observed that the policy trained using Rc-RM exhibits high semantic quality, which significantly surpasses other competing methods, further demonstrating the effectiveness of our method.

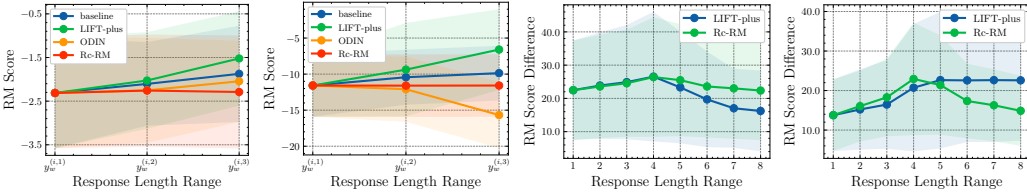

(a) Qwen2-1.5B-Instruct  (b) Llama-3.1-8B-Instruct  (c) Qwen2-1.5B-Instruct  (d) Llama-3.1-8B-Instruct

Figure 3: (a), (b): Analysis of reward scores across models on $\mathcal{D}_{eval}^{ml}$ as a function of response length, with smaller changes indicating reduced length bias. (c), (d): Comparison of score differences between LIFT-plus and Rc-RM on $\mathcal{D}_{eval}^{mls}$ under varying *word_num* constraints. An ideal reward model should show an initial increase in score difference followed by a return to its initial value.

Furthermore, for $\mathcal{D}_{eval}$, we rewrite each $(x^{(i)}, y_c^{(i)})$ into multiple length-varied pairs $\mathcal{D}_{eval}^{ml} = \{(x^{(i)}, y_c^{(i,j)})\}, j \in \{1, 2, 3\}$, where $|y_c^{(i,1)}| < |y_c^{(i,2)}| < |y_c^{(i,3)}|$, while maintaining the semantic consistency between $y_c^{(i)}$ and $y_c^{(i,j)}$. Then we evaluate each reward model on $\mathcal{D}_{eval}^{ml}$, with the results shown in Figure 3, where the reward model with less length bias exhibits smaller slopes in its scores. The results indicate that both Baseline and LIFT-plus show an upward trend in their scores, while ODIN fluctuates due to varying degrees of length penalty. Rc-RM, however, demonstrates superior stability, clearly highlighting its effectiveness in mitigating length bias.

## 5.3 THE RESULTS OF DPO MODELS

**Rc-DPO effectively reduces response length and improves quality.** Table 5 highlights the effectiveness of Rc-DPO in mitigating length bias and enhancing response quality on AlpacaEval. Specifically, LIFT-plus negatively impacts semantic quality, leading to a notable decline in *Quality Win Ratio*. Compared to R-DPO, Dr. DPO, SimPO, and CPO, on Qwen2.5-7B-Base, Rc-DPO achieves a *Quality Win Ratio* of 45.39%, outperforming the Baseline by 11.85% and R-DPO by 4.99%, while reducing the response length from 517.30 to 208.42. Similarly, on Llama-3.1-8B-Instruct, Rc-DPO attains the highest *Quality Win Ratio* of 64.34%, surpassing SimPO by 6.21% and Dr. DPO by 12.16%. These results demonstrate that Rc-DPO consistently achieves the best trade-off between response quality and length control across both Base and Instruct models, further validating the effectiveness and generalizability of our approach.

Table 5: Evaluation results of different DPO models on AlpacaEval (Dubois et al., 2024b).

| Metrics | Qwen2.5-7B-Base | | | | | | |
| --- | --- | --- | --- | --- | --- | --- | --- |
| | Baseline | LIFT-plus | R-DPO | Dr.DPO | SimPO | CPO | Rc-DPO |
| Quality Win Ratio (%) | 33.54 | 31.67 | 40.40 | 39.74 | 41.26 | 40.07 | **45.39** |
| Response Length | 517.30 | 184.17 | 583.18 | 311.49 | 286.54 | 254.86 | 208.42 |
| Metrics | Qwen2.5-7B-Instruct | | | | | | |
| | Baseline | LIFT-plus | R-DPO | Dr.DPO | SimPO | CPO | Rc-DPO |
| Quality Win Ratio (%) | 28.43 | 25.69 | 34.16 | 36.12 | 37.16 | 35.72 | **44.63** |
| Response Length | 261.32 | 195.80 | 235.39 | 247.19 | 281.26 | 257.95 | 228.24 |
| Metrics | Llama-3.1-8B-Base | | | | | | |
| | Baseline | LIFT-plus | R-DPO | Dr.DPO | SimPO | CPO | Rc-DPO |
| Quality Win Ratio (%) | 46.30 | 40.15 | 49.13 | 51.12 | 54.38 | 50.86 | **58.10** |
| Response Length | 435.98 | 157.80 | 465.72 | 347.29 | 309.86 | 288.64 | 202.01 |
| Metrics | Llama-3.1-8B-Instruct | | | | | | |
| | Baseline | LIFT-plus | R-DPO | Dr.DPO | SimPO | CPO | Rc-DPO |
| Quality Win Ratio (%) | 42.52 | 47.88 | 42.64 | 52.18 | 58.13 | 54.67 | **64.34** |
| Response Length | 247.74 | 153.77 | 215.82 | 229.17 | 218.46 | 244.04 | 204.77 |

Table 6: Evaluation results of Rc-RM $w/o$ $\mathcal{D}_{Rc}^c$ and $w/o$ $\mathcal{D}_{Rc}^r$ on $\mathcal{D}_{eval}^q$ and $\mathcal{D}_{eval}^l$.

| Metrics | Qwen2-1.5B-Base | | Qwen2-1.5B-Instruct | |
| --- | --- | --- | --- | --- |
| | $w/o\ \mathcal{D}_{Rc}^c$ | $w/o\ \mathcal{D}_{Rc}^r$ | $w/o\ \mathcal{D}_{Rc}^c$ | $w/o\ \mathcal{D}_{Rc}^r$ |
| Quality Eval Acc (%) | 58.78 | 55.59 | 59.31 | 57.71 |
| Length Eval Acc (%) | 45.83 | 36.22 | 45.51 | 44.87 |
| Metrics | Llama-3.1-8B-Base | | Llama-3.1-8B-Instruct | |
| | $w/o\ \mathcal{D}_{Rc}^c$ | $w/o\ \mathcal{D}_{Rc}^r$ | $w/o\ \mathcal{D}_{Rc}^c$ | $w/o\ \mathcal{D}_{Rc}^r$ |
| Quality Eval Acc (%) | 50.27 | 50.53 | 58.51 | 52.13 |
| Length Eval Acc (%) | 44.87 | 32.69 | 37.18 | 33.65 |

## 5.4 ABLATION STUDIES

Due to the high cost of evaluating the DPO model, we focus on the reward model for ablation studies. For convenience, we denote $\{(x, x_l^1, y_w)\}$ as $\mathcal{D}_{Rc}^c$ and $\{(x_l^2, x, y_l)\}$ as $\mathcal{D}_{Rc}^r$. We report the key ablation results in this section, with extended experiments provided in Appendix F.7.

$\mathcal{D}_{Rc}^c$ **and** $\mathcal{D}_{Rc}^r$ **are complementary.** We conduct ablation experiments by training with $\mathcal{D}_{rm} \cup \mathcal{D}_{Rc}^c$ ($w/o\ \mathcal{D}_{Rc}^r$) and $\mathcal{D}_{rm} \cup \mathcal{D}_{Rc}^r$ ($w/o\ \mathcal{D}_{Rc}^c$), with the results shown in Table 6. As observed, when only $\mathcal{D}_{Rc}^c$ or $\mathcal{D}_{Rc}^r$ is used, the *Quality Eval Acc* of Rc-RM drops significantly, nearing the performance of Baseline in Table 4, while its *Length Eval Acc* hovers around $50\%$, failing to learn length instructions. Subsequently, we prepend a length constraint to each $x$ in $\mathcal{D}_{eval}^q$, forming $x_l$, where $y_c$ satisfies the length constraint, resulting in $\mathcal{D}_{eval}^{q,l}$. The evaluation results on both $\mathcal{D}_{eval}^q$ and $\mathcal{D}_{eval}^{q,l}$, shown in Figure 8. Specifically, Figure 8(a) indicates that for the reward models trained with $\mathcal{D}_{rm} \cup \mathcal{D}_{Rc}^c$, the scores of $(x_l, y_c)$ are consistently lower than those of the original $(x, y_c)$, despite $y_c$'s length satisfies $x_l$. A similar phenomenon is observed in $\mathcal{D}_{rm} \cup \mathcal{D}_{Rc}^r$ (Figure 8(b)). Therefore, combining $\mathcal{D}_{Rc}^c$ and $\mathcal{D}_{Rc}^r$ is essential for preventing the model from developing new biases.

## 6 CONCLUSION

We propose Rc-BT, a method that separates human semantic intent from length instructions to mitigate length bias while retaining length sensitivity, and provide mathematical extensions to RM and DPO. Extensive experiments demonstrate the effectiveness of Rc-BT in reducing length bias and improving length instruction following, with potential for broader bias mitigation.

ACKNOWLEDGEMENT

This work was supported by the National Natural Science Foundation of China under Contract 623B2097, the Youth Innovation Promotion Association CAS. It was also supported by Merchants Union Consumer Finance Company Limited, and the GPU cluster built by MCC Lab of USTC & the Supercomputing Center of USTC

REPRODUCIBILITY STATEMENT

We have made significant efforts to ensure the reproducibility of our work. For theoretical contributions, Section 4 and Appendix C provides clear explanations and complete mathematical derivations of our proposed Rc-BT framework. Regarding the experimental setup, Section 3 and 5, Appendices D and F contain a full description of the data processing procedures used to construct training and evaluation datasets. Together, these materials are intended to facilitate accurate reproduction and verification of our results.

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

# A  SUMMARY OF NOTATIONS

Table 7 provides a summary of the notations used throughout this paper. The table is intended to serve as a quick reference to help readers understand the mathematical symbols and datasets introduced in the paper.

Table 7: List of notations and their definitions used throughout the paper

| Notations | Description |
|---|---|
| $x$ | The original prompt. |
| $x_e$ | The "empty prompt" string. |
| $x_l$ | Prompt $x$ augmented with a length constraint. |
| $x_l^i$ | Variant of $x_l$ with a specific length constraint. |
| $x_c$ | Chosen prompt for $y$. |
| $x_r$ | Rejected prompt for $y$. |
| $y_c$ | Chosen response for $x$. |
| $y_c^i$ | Semantically equivalent rewrite of $y_c$. |
| $y_r$ | Rejected response for $x$. |
| $y_r^i$ | Semantically equivalent rewrite of $y_r$. |
| $\mathcal{D}_{sft}$ | Training Dataset used for SFT. |
| $\mathcal{D}_{rm}$ | Training Dataset used for reward modeling and DPO. |
| $\mathcal{D}_{Rc}$ | Augmented training dataset for Rc-BT based on $\mathcal{D}_{rm}$. |
| $\mathcal{D}_{Rc}^c$ | Subset of $\mathcal{D}_{Rc}$ containing $y_c$. |
| $\mathcal{D}_{Rc}^r$ | Subset of $\mathcal{D}_{Rc}$ containing $y_r$. |
| $\mathcal{D}_{eval}$ | Original evaluation dataset for the reward model. |
| $\mathcal{D}_{eval}^e$ | Derived from $\mathcal{D}_{eval}$ by replacing all prompts with $x_e$ (empty evaluation dataset). |
| $\mathcal{D}_{eval}^r$ | Derived from $\mathcal{D}_{eval}$ by randomly matching prompts with responses from other samples (random evaluation dataset). |
| $\mathcal{D}_{eval}^q$ | Evaluation dataset for assessing semantic quality of the reward model (quality evaluation dataset). |
| $\mathcal{D}_{eval}^l$ | Evaluation dataset for assessing length-following ability of the reward model (length evaluation dataset). |
| $r_\phi$ | Reward model with parameters $\phi$. |
| $\pi_{SFT}$ | Initialization LLM. |
| $\pi_\theta$ | LLM with parameters $\theta$. |
| $\pi_{ref}$ | Reference model used in GRPO/PPO/DPO. |

# B  RELATED WORK

**Reinforcement Learning from Human Feedback.** Reinforcement Learning from Human Feedback (RLHF) (Christiano et al., 2017; Ziegler et al., 2019) has established itself as a dominant paradigm for aligning LLMs with human preferences (Achiam et al., 2023; Team et al., 2023; Caruccio et al., 2024; Yang et al., 2024; Dubey et al., 2024). At its core, RLHF trains models by optimizing reward signals derived from human feedback, typically collected through preference comparisons or explicit ratings (Bai et al., 2022b; Lee et al., 2023). This approach has demonstrated significant efficacy in refining LLM behavior to align with human expectations across diverse complex tasks (Kreutzer et al., 2018; Liu et al., 2020; Ziegler et al., 2019; Ouyang et al., 2022).

However, RLHF necessitates training a reward model (RM) to approximate human preferences, followed by LLM alignment through reinforcement learning (RL) algorithms, notably Proximal Policy Optimization (PPO) (Schulman et al., 2017). This process presents significant challenges and stability issues (Queeney et al., 2021; Gu et al., 2022). Direct preference alignment methods, in contrast, provide a more robust training approach (Ivison et al., 2024; Hong et al., 2024; Xu et al., 2024b). Among these methods, direct preference optimization (DPO) (Rafailov et al., 2024) has emerged as a leading technique, inspiring various derivative algorithms (Hong et al., 2024; Chen

et al., 2024a; Ethayarajh et al., 2024). By reformulating the RLHF reward function, DPO eliminates both the need for an online reward model and the instabilities inherent in RL algorithms, enabling stable offline preference learning.

Our method enhances RLHF by improving reward model robustness and seamlessly integrates into DPO, enabling more effective LLM training.

**Reward Hacking.** Despite its promising performance, RLHF is vulnerable to reward hacking — the over-optimization of the reward model (Skalse et al., 2022; Pan et al., 2022; Casper et al., 2023; Lambert & Calandra, 2023). This phenomenon occurs when the policy exploits inherent biases in the reward model to maximize rewards without achieving intended objectives. Similar exploitation patterns have been observed in DPO-trained LLMs (Lambert et al., 2024; Xu et al., 2024c). The root causes are multifaceted: task complexity and subjectivity (Casper et al., 2023), evaluation criteria limitations (Parrish et al., 2022), and evaluator qualification constraints (Skalse et al., 2022). Consequently, human preference data exhibit biases and inconsistencies (Dubois et al., 2024b), and reward models struggle with preference approximation and out-of-distribution (OOD) generalization.

One of the most common forms of reward hacking is length bias (Shen et al., 2023; Singhal et al., 2023; Park et al., 2024; Chen et al., 2024b), where the reward models tend to favor longer responses, assigning higher reward scores to longer responses even if their semantic contents are not of high quality. This tendency is largely driven by human evaluators' preference for longer answers, which is easily exploited by the LLMs. As a result, LLMs often generate unnecessarily lengthy replies to appear more detailed or better formatted, even when the actual quality remains unchanged. To address this issue, several methods have been proposed. ODIN (Chen et al., 2024b) adds multiple length regularization terms to the RM's training objective, which helps to distinguish response quality from length and effectively controls the LLM's dependency on response length. Shen et al. (2023) involve using dual-reward model and different sets of learning rate hyperparameters, allowing different part of the reward model to learn distinct paradigms, thereby removing length bias from final reward scores. Length Regularized DPO (R-DPO) (Park et al., 2024) modify the DPO training objective by adding a length penalty term to prevent length bias exploitation.

However, the above methods assume that the model's sensitivity to length information is detrimental and, therefore, seek to eliminate the model's perception of length information. In contrast, our approach enables the model to explicitly distinguish between human semantic intent and length instruction, preserving its ability to perceive response length rather than completely eliminating the perception of length information.

**Length Instruction Following.** Current state-of-the-art (SOTA) LLMs, both open-source and closed-source, demonstrate a certain degree of implicit ability to follow length instructions (Yuan et al., 2024). For example, adding terms like "concise" or "verbose" can influence the length of the model's outputs. However, when it comes to explicit length instructions, such as "Answer the following instruction using 150 words or less.", these models often fail to adhere effectively. To address this, LIFT (Yuan et al., 2024) enhances the model's length instruction following capability by constructing an explicit length instruction preference dataset. However, the dataset used by LIFT is built at the expense of semantic quality, resulting in a degradation of the model's output quality. In contrast, our model not only effectively learns length instructions but also maintains, even in many cases improves, the semantic quality of the model's outputs.

## C  DERIVATION OF THE DPO OBJECTIVE UNDER THE RESPONSE-CONDITIONED BRADLEY-TERRY MODEL

**Derivation of RL Fine-Tuning Objective.** During the RL phase of traditional RLHF, the trained reward model $r_\phi$ serves as the feedback mechanism for policy optimization. The standard RL optimization objective is formulated as:

$$\max_{\pi_\theta} \mathbb{E}_{x\sim\mathcal{D}, y\sim\pi_\theta(y|x)}\big[r_\phi(x,y)\big] - \beta\mathbb{D}_{\text{KL}}\big[\pi_\theta(y\mid x)\mid\mid\pi_{\text{ref}}(y\mid x)\big], \qquad (9)$$

under the reference policy $\pi_{\text{ref}}$ and the parametrized policy $\pi_\theta$. In the response-conditioned scenario, where responses are predetermined and guide prompt selection, the KL-divergence constraint

in Eqn. 9 requires modification. Accordingly, we reformulate the response-conditioned RL optimization objective as:

$$\max_{\pi_\theta} \mathbb{E}_{y \sim \mathcal{D}_{Rc}, x \sim \pi_\theta(x|y)} \big[ r_\phi(x,y) \big] - \beta \mathbb{D}_{KL} \big[ \pi_\theta(x \mid y) \,||\, \pi_{\text{ref}}(x \mid y) \big]. \tag{10}$$

For simplicity, we unify the notation of $\mathcal{D}_{Rc} = \{(x, x_l^1, y_c)\} \cup \{(x_l^2, x, y_r)\}$ as $\{(x_c, x_r, y)\}$. Next, similar to the derivation in DPO (Park et al., 2024), we can derive the partition function $Z(y)$ from Eqn. 10:

$$Z(y) = \sum_x \pi_{\text{ref}}(x \mid y) \exp \left( \frac{1}{\beta} r_\phi(x,y) \right). \tag{11}$$

Since the partition function is a function only of the response $y$ and the reference policy $\pi_{\text{ref}}$, and does not depend on the prompt $x$ or the parametrized policy $\pi_\theta$ to be optimized, we can reorganize Eqn. 10 to obtain the following objective:

$$\min_{\pi_\theta} \mathbb{E}_{y \sim \mathcal{D}} \left[ \mathbb{E}_{x \sim \pi_\theta(x|y)} \left[ \log \frac{\pi_\theta(x|y)}{\pi^*(x|y)} \right] - \log Z(y) \right] = \min_{\pi_\theta} \mathbb{E}_{y \sim \mathcal{D}} \left[ \mathbb{D}_{KL}(\pi_\theta(x|y) \,||\, \pi^*(x|y)) - \log Z(y) \right], \tag{12}$$

$$\pi^*(x \mid y) = \frac{1}{Z(y)} \pi_{\text{ref}}(x \mid y) \exp \left( \frac{1}{\beta} r_\phi(x,y) \right), \tag{13}$$

where $\pi^*(x \mid y)$ is a valid probability distribution which $\pi^*(x \mid y) > 0$ and $\sum_x \pi^*(x \mid y) = 1$. By using Gibbs' inequality, the KL divergence is minimized to $0$ if and only if the two distributions are identical. Therefore, we obtain the optimal solution:

$$\pi_\theta(x \mid y) = \pi^*(x \mid y) = \frac{1}{Z(y)} \pi_{\text{ref}}(x \mid y) \exp \left( \frac{1}{\beta} r_\phi(x,y) \right). \tag{14}$$

Finally, by taking the logarithm on both sides and performing some basic algebraic operations, we can express the reward model $r_\phi(x,y)$ through its corresponding optimal policy $\pi^*(x \mid y)$:

$$r_\phi(x,y) = \beta \log \frac{\pi^*(x \mid y)}{\pi_{\text{ref}}(x \mid y)} + \beta \log Z(y). \tag{15}$$

**Derivation of Response-conditioned DPO Objective.** Following Section 4.2, the Response-conditioned Bradley-Terry model is formulated as:

$$p^*(x_c \succ x_r | y) = \frac{\exp\left(r^*(x_c, y)\right)}{\exp(r^*(x_c, y)) + \exp(r^*(x_r, y))}. \tag{16}$$

In the above derivation, we demonstrated that the optimal reward $r^*(x,y)$, parameterized as $r_\phi(x,y)$, can be expressed in term of its corresponding optimal policy $\pi^*(x \mid y)$. By substituting Eqn. 15 into Eqn. 16 and preforming simple simplifications, we obtain:

$$p^*(x_c \succ x_r|y) = \frac{\exp\left(\beta \log \frac{\pi^*(x_c|y)}{\pi_{\text{ref}}(x_c|y)} + \beta \log Z(y)\right)}{\exp\left(\beta \log \frac{\pi^*(x_c|y)}{\pi_{\text{ref}}(x_c|y)} + \beta \log Z(y)\right) + \exp\left(\beta \log \frac{\pi^*(x_r|y)}{\pi_{\text{ref}}(x_r|y)} + \beta \log Z(y)\right)}$$

$$= \frac{1}{1 + \exp\left(\beta \log \frac{\pi^*(x_r|y)}{\pi_{\text{ref}}(x_r|y)} - \beta \log \frac{\pi^*(x_c|y)}{\pi_{\text{ref}}(x_c|y)} + (\beta \log Z(y) - \beta \log Z(y))\right)}$$

$$= \frac{1}{1 + \exp\left(\beta \log \frac{\pi^*(x_r|y)}{\pi_{\text{ref}}(x_r|y)} - \beta \log \frac{\pi^*(x_c|y)}{\pi_{\text{ref}}(x_c|y)}\right)}$$

$$= \sigma\left(\beta \log \frac{\pi^*(x_c|y)}{\pi_{\text{ref}}(x_c|y)} - \beta \log \frac{\pi^*(x_r|y)}{\pi_{\text{ref}}(x_r|y)}\right). \tag{17}$$

Since we cannot directly access $\pi(x|y)$, we reparameterize it using Bayes' theorem for conditional probability decomposition, transforming Eqn. 17 into a joint probability formulation as follows:

$$p^*(x_c \succ x_r|y) = \sigma\left(\beta \log \frac{\pi^*(x_c|y)}{\pi_{\text{ref}}(x_c|y)} - \beta \log \frac{\pi^*(x_r|y)}{\pi_{\text{ref}}(x_r|y)}\right)$$

$$= \sigma\left(\beta \log \frac{\frac{\pi^*(x_c,y)}{\pi^*(y)}}{\frac{\pi_{\text{ref}}(x_c,y)}{\pi_{\text{ref}}(y)}} - \beta \log \frac{\frac{\pi^*(x_r,y)}{\pi^*(y)}}{\frac{\pi_{\text{ref}}(x_r,y)}{\pi_{\text{ref}}(y)}}\right)$$

$$= \sigma\left(\beta \log \frac{\pi^*(x_c, y)}{\pi_{\text{ref}}(x_c, y)} - \beta \log \frac{\pi^*(x_r, y)}{\pi_{\text{ref}}(x_r, y)}\right). \tag{18}$$

Finally, we derive the loss function for Rc-DPO for a parameterized policy $\pi_\theta$ using the response-conditioned preference dataset $\mathcal{D}_{Rc} = \{(x_c, x_r, y)\}$ by applying maximum likelihood estimation:

$$\mathcal{L}_{DPO}^{Rc}(\pi_\theta; \pi_{\text{ref}}) = -\mathbb{E}_{(x_c,x_r,y)\sim\mathcal{D}_{Rc}}\left[\log \sigma\left(\beta \log \frac{\pi_\theta(x_c, y)}{\pi_{\text{ref}}(x_c, y)} - \beta \log \frac{\pi_\theta(x_r, y)}{\pi_{\text{ref}}(x_r, y)}\right)\right]. \tag{19}$$

## D   THE IMPLEMENTATION DETAILS OF PRELIMINARY EXPLORATIONS

### D.1   TRAINING DATASET AND MODELS

**Training Dataset.** Similar to the data processing approach used by Yuan et al. (2024), we extract the first turn of English conversations from the OpenAssistant dataset (Köpf et al., 2024) and use it as our complete dataset. Based on their human annotated ranking, we label rank 0 as "chosen" ($y_c$) and rank 1 as "rejected" ($y_r$), resulting in the complete dataset $\mathcal{D}$, that each example is a triple $(x^{(i)}, y_c^{(i)}, y_r^{(i)})$.

For training of different models and subsequent experimental analysis, we first divide $\mathcal{D}$ into a training dataset $\mathcal{D}_{train}$, which contains 90% of the data for model training, and an evaluation dataset $\mathcal{D}_{eval}$, comprising the remaining 10% for model evaluation. Next, we further divide $\mathcal{D}_{train}$ into two subsets: a supervised fine-tuning (SFT) training dataset $\mathcal{D}_{sft}$, containing 30% of $\mathcal{D}_{train}$, and a reward model (RM) training dataset $\mathcal{D}_{rm}$, containing the remaining 70% of $\mathcal{D}_{train}$. For all reward models, we incorporate $\mathcal{D}_{rm}$ into the training process to ensure the fundamental semantic comprehension ability of the trained reward models. For example, in Section 3.4, each reward model is trained using the variant dataset (LIFT-plus$_2^{reverse}$ / LIFT-plus$_2^{noreverse}$ / LIFT-plus$_2^{empty}$) combined with $\mathcal{D}_{rm}$.

For the *Base* models, we first apply SFT using $\mathcal{D}_{sft}$, followed by reward model training on $\mathcal{D}_{rm}$, For the *Instruct* models, we directly train the reward model using $\mathcal{D}_{rm}$ without prior supervised fine-tuning.

**Training Models.** In the preliminary explorations, we employ three groups of models — Qwen2-1.5B-Base with Qwen2-1.5B-Instruct, Qwen2.5-7B-Base with Qwen2.5-7B-Instruct, and Llama-3.1-8B-Base with Llama-3.1-8B-Instruct — to verify the broad applicability of our analytical results. However, in Section 3.4, we exclude the Qwen2.5-7B-Base and Qwen2.5-7B-Instruct models from our analysis for simplicity.

## D.2 THE CONSTRUCTION PROCESS OF $\mathcal{D}_{eval}^q$

To fairly and accurately evaluate the semantic comprehension capabilities of the reward model, we establish a refined quality evaluation dataset $\mathcal{D}_{eval}^q$ through an automated transformation of $\mathcal{D}_{eval}$, leveraging advanced language models (Dubois et al., 2024b; Zheng et al., 2023; Chiang et al., 2023). To ensure the reliability of the generated dataset, we did not directly adopt GPT-4o's outputs, as GPT-4o also exhibits length bias. Instead, we employed a rigorous algorithm that augments simple model generation with programmatic verification, thereby transforming the process into generation + verification.

Specifically, First, for each triplet $(x^{(i)}, y_c^{(i)}, y_r^{(i)})$ in $\mathcal{D}_{eval}$, we employ GPT-4o (Hurst et al., 2024) with the *Response Rewriting Template* (Figure 4(a)) to generate two alternative responses, $y_c^{(i),1}$ and $y_c^{(i),2}$, based on the original prompt $x^{(i)}$ and response $y_c^{(i)}$. The semantic equivalence between the generated response $y_c^{(i),1}$ (or $y_c^{(i),2}$) and the original response $y_c^{(i)}$ is verified using the *Quality Consistency Verification Template* in Figure 4(d).

Subsequently, we process $y_r^{(i)}$ through GPT-4o to generate two alternative responses, $y_r^{(i),1}$ and $y_r^{(i),2}$, based on the original prompt $x^{(i)}$. These generated responses must satisfy two criteria: (1) maintain semantic equivalence with $y_r^{(i)}$, verified using the *Quality Consistency Verification Template*, and (2) exhibit specific length relationships with their counterparts, namely $|y_r^{(i),1}| > |y_c^{(i),1}|$ and $|y_r^{(i),2}| < |y_c^{(i),2}|$, verified using programmatic evaluation. Given GPT-4o's limitations in direct length control, we implement a separate generation process utilizing the *Response Expansion Template* and *Response Compression Template* (Figure 4(b) and 4(c)) to obtain $y_r^{(i),1}$ and $y_r^{(i),2}$, respectively.

The resulting quality evaluation dataset is formulated as $\mathcal{D}_{eval}^q = \{(x^{(i)}, y_c^{(i),1}, y_r^{(i),1})\} \cup \{(x^{(i)}, y_c^{(i),2}, y_r^{(i),2})\}$. The complete prompting templates for response generation and quality consistency verification are presented in Figure 4.

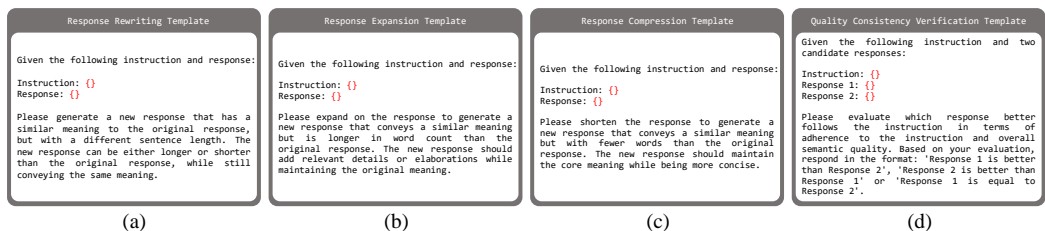

Figure 4: **(a).** Response rewriting template for the chosen response $y_c^{(i)}$; **(b).** Response expansion template for the rejected response $y_r^{(i)}$; **(c).** Response compression template for the rejected response $y_r^{(i)}$; **(d).** Quality consistency verification template for assessing the quality consistency between the rewritten and the original responses.

## D.3 THE CONSTRUCTION PROCESS OF $\mathcal{D}_{eval}^l$

To accurately and efficiently evaluate the reward model's adherence to length instructions, we establish the length evaluation dataset $\mathcal{D}_{eval}^l$. Each triplet $(x_l^{(i)}, y_c^{(i),l}, y_r^{(i),l})$ in $\mathcal{D}_{eval}^l$ consists of a length instruction $x_l^{(i)} = length\ constraint + x^{(i)}$ and two semantically equivalent responses that differ only

in length: $y_c^{(i),l}$ adheres to the length constraint specified in $x_l^{(i)}$, while $y_r^{(i),l}$ violates it[8]. To optimize computational efficiency, we derive $\mathcal{D}_{eval}^l$ from the previously constructed $\mathcal{D}_{eval}^q$. Specifically, for the responses in $\mathcal{D}_{eval}^q$, we already ensure that $y_c^{(i),1}$ and $y_c^{(i),2}$ maintain semantic similarity during the construction of $\mathcal{D}_{eval}^q$. Additionally, the stochastic nature of GPT-4o's response generation inherently produces responses of varying lengths, with $|y_c^{(i),1}|$ and $|y_c^{(i),2}|$ typically differing. Leveraging this property, we select these responses ($y_c^{(i),1}$ and $y_c^{(i),2}$) as candidates for $x_l^{(i)}$. Specifically, we set the *word_num* parameter in $x_l^{(i)}$ within the range $[|y_c^{(i),1}|, |y_c^{(i),2}|]$ (assuming $|y_c^{(i),1}| < |y_c^{(i),2}|$, $x_l^{(i)} = less\ length\ constraint + x^{(i)}$). Consequently, we assign $y_c^{(i),l} = y_c^{(i),1}$ as the chosen response and $y_r^{(i),l} = y_c^{(i),2}$ as the rejected response. Finally, the constructed length evaluation dataset is $\mathcal{D}_{eval}^l = \{(x_l^{(i)}, y_c^{(i),l}, y_r^{(i),l})\}$.

## D.4 THE EXTENSION PROCESS OF LIFT

In LIFT (Yuan et al., 2024), the authors propose a straightforward method for constructing a length instruction dataset. In Section 3.4, we will examine the limitations of this method. The LIFT approach can be divided into two types: the first part $\text{LIFT}_1$ constructs length instruction $x_l^{(i)}$ that both responses ($y_c^{(i)}$ and $y_r^{(i)}$) satisfy the length constraint specified in $x_l^{(i)}$; the second part $\text{LIFT}_2$ constructs length instruction $x_l^{(i)}$ that only one response ($y_c^{(i)}$ or $y_r^{(i)}$) satisfies. In cases where the rejected response $y_r^{(i)}$ adheres to the length instruction but the chosen response $y_c^{(i)}$ does not, this can lead to a reversal of the original preference order, which fundamentally undermines the semantic quality of the data. Furthermore, LIFT only considers a maximum length constraint, i.e., *"less length constraint"*, without accounting for a minimum length constraint, i.e., *"more length constraint"*. Therefore, we made a slight improvement to LIFT by adding an *"more length constraint"*, resulting in LIFT-plus:

$$
\begin{aligned}
\text{LIFT-plus}_{less} &= \text{LIFT} = \text{LIFT}_1 \cup \text{LIFT}_2 \\
&= \{(x_{l,less}^{(i)}, y_c^{(i)}, y_r^{(i)})\} \cup \{(x_{l,less}^{(i)}, y_r^{(i)}, y_c^{(i)})\}, \\
x_{l,less}^{(i)} &= less\ length\ constraint + x^{(i)}, \\
\text{LIFT-plus}_{more} &= \{(x_{l,more}^{(i)}, y_c^{(i)}, y_r^{(i)})\} \cup \{(x_{l,more}^{(i)}, y_r^{(i)}, y_c^{(i)})\}, \\
x_{l,more}^{(i)} &= more\ length\ constraint + x^{(i)}, \\
\text{LIFT-plus} &= \text{LIFT-plus}_{less} \cup \text{LIFT-plus}_{more}.
\end{aligned}
\tag{20}
$$

where less length constraint is "Answer the following instruction using $\{word\_num\}$ words or less.", and more length constraint is "Answer the following instruction using $\{word\_num\}$ words or more." In this article, we refer to $x_{l,less}^{(i)}$ as the "or less" length instruction and $x_{l,more}^{(i)}$ as the "or more" length instruction. Similarly, LIFT-plus can also be decomposed into two parts: $\text{LIFT-plus}_1$ and $\text{LIFT-plus}_2$ as follows:

$$
\begin{aligned}
\text{LIFT-plus} = \text{LIFT-plus}_1 \cup \text{LIFT-plus}_2 &= \{(x_l^{(i)}, y_c^{(i)}, y_r^{(i)})\} \cup \{(x_l^{(i)}, y_r^{(i)}, y_c^{(i)})\}, \\
x_l^{(i)} &= length\ constraint + x^{(i)}.
\end{aligned}
\tag{21}
$$

where length constraint is less length constraint or more length constraint. In the first part $\text{LIFT-plus}_1$, the length instruction $x_l^{(i)}$ is satisfied to both responses. As a result, it may degrade into a length-agnostic instruction, where the model disregards the added length constraint and focuses solely on the original prompt $x^{(i)}$. Therefore, the key component that effectively facilitates adherence to length instructions is the second part, $\text{LIFT-plus}_2$, where only one response ($y_c^{(i)}$ or $y_r^{(i)}$) satisfies $x_l^{(i)}$.

---

[8]The detailed definition of the *length constraint* can be found in Appendix D.4.

Furthermore, we divide LIFT-plus$_2$ into two subsets: LIFT-plus$_2^{reverse} = \{(x_l^{(i)}, y_r^{(i)}, y_c^{(i)})\}$, which reverses the original preference order between chosen and rejected responses due to the length constraint in $x_l^{(i)}$, and LIFT-plus$_2^{noreverse} = \{(x_l^{(i)}, y_c^{(i)}, y_r^{(i)})\}$, which preserves the original preference order. In addition, based on LIFT-plus$_2^{noreverse}$, we construct a dataset purely centered on length instructions, denoted as LIFT-plus$_2^{empty} = (x_e^{(i)}, y_c^{(i)}, y_r^{(i)})$, where $x_e^{(i)} = $ *length constraint* + *["empty prompt"]*. This dataset isolates length instructions from other semantic content and only focuses on the length instructions themselves. Here, LIFT-plus$_2^{empty}$ is specifically designed to investigate the impact of semantic prompts on length instruction following in Section 3.4.

# E    THE DETAILED RESULTS AND ANALYSIS OF PRELIMINARY EXPLORATIONS

## E.1    LENGTH BIAS INDEED EXISTS

Although the responses $y_c^{(i)}$ and $y_r^{(i)}$ in $\mathcal{D}_{eval}^e$, $\mathcal{D}_{eval}^r$, and $\mathcal{D}_{eval}^{s,r}$ are semantically misaligned with their respective prompt, we observe that the reward models still achieve relatively high accuracies on these evaluation datasets. Specifically, the results for the three groups of models on $\mathcal{D}_{eval}^e$, $\mathcal{D}_{eval}^r$, and $\mathcal{D}_{eval}^{s,r}$, as shown in Table 1, indicate that nearly all reward models achieve an accuracy of $60\%$ or higher, which is very close to the accuracy observed on the original evaluation dataset $\mathcal{D}_{eval}$. This suggests that, even in the absence of the prompt, the reward models exhibit a significant bias toward the chosen responses.

To further investigate whether this bias primarily arises from response length, we first analyze the consistency of model evaluation results between $\mathcal{D}_{eval}^e$ (or $\mathcal{D}_{eval}^r$) and $\mathcal{D}_{eval}$, focusing on how varying prompts affect the selection of the same response. The results, presented in Table 1, show that in over $85\%$ of cases, the models select the same response regardless of the prompt. This reinforces the observation that the reward models' preference is not primarily driven by the prompts themselves but rather by other response-related factors.

Next, we plot the relationship between the response length and the corresponding reward score of the reward models for $\mathcal{D}_{eval}$, $\mathcal{D}_{eval}^e$, $\mathcal{D}_{eval}^r$, and $\mathcal{D}_{eval}^{s,r}$, as shown in Figure 1. To facilitate comparison, we normalize the reward scores of both models (Qwen2-1.5B-Instruct and Llama-3.1-8B-Instruct) to the same range. The results reveal a strong linear correlation between the response length and the reward score across all three evaluation datasets. Specifically, as response length increases, the model's reward score also rises, indicating that response length plays a significant role in the model's assessment of response quality.

We further conducted experiments that analyzing the relationship between the accuracy of reward models on $\mathcal{D}_{eval}$ and the length difference between the chosen and rejected responses. The results are demonstrated in Table 8. It can be observed that as the length difference increases, the accuracy also increases. This clearly demonstrates the presence of length bias in the Baseline models.

Table 8: Accuracy of different reward models on different subsets of $\mathcal{D}_{eval}$ partitioned by length difference between chosen and rejected responses

| Model | $\leq$ -100 | -100 $\sim$ -50 | -50 $\sim$ 0 | 0 $\sim$ 50 | 50 $\sim$ 100 | $\geq$ 100 |
|---|---|---|---|---|---|---|
| Qwen2-1.5B-Instruct | 9.10% | 6.06% | 38.96% | 84.40% | 89.79% | 95.52% |
| Qwen2.5-7B-Instruct | 18.19% | 15.16% | 29.87% | 81.65% | 91.84% | 88.46% |
| Llama-3.1-8B-Instruct | 18.19% | 9.10% | 40.26% | 67.89% | 79.59% | 88.06% |

## E.2    LENGTH BIAS IN EVALUATION DATASET

We conducted further analysis on the generated evaluation dataset $\mathcal{D}_{eval}^q$. Specifically, we recruited 20 volunteers, each assigned 10 randomly sampled data from $\mathcal{D}_{eval}^q$. The volunteers were instructed to focus solely on the semantic quality of the responses, disregarding factors such as length or

format. We then collected their preferences regarding the data. Upon aggregating the results, we found that human preferences are 90.5% consistent with $\mathcal{D}^q_{eval}$. This strongly supports the validity of our generated datasets.

### E.3 LENGTH INSTRUCTIONS ARE EASILY LEARNED

The results for the Qwen2-1.5B and Llama-3.1-8B models trained on the three variant datasets (LIFT-plus$_2^{reverse}$, LIFT-plus$_2^{noreverse}$, and LIFT-plus$_2^{empty}$) are presented in Table 9. As observed, most reward models achieve similar or even lower accuracy on $\mathcal{D}^q_{eval}$ compared to Baseline models in Table 4, while their accuracy on $\mathcal{D}^l_{eval}$ remains relatively high. These results suggest that, although reward models are trained on diverse length instruction datasets, they all primarily learn to capture response length information rather than balancing adherence to length instructions with attention to the semantic content of the prompt.

Table 9: Evaluation results of different reward models (LIFT-plus variants) on quality ($\mathcal{D}^q_{eval}$) and length ($\mathcal{D}^l_{eval}$) evaluation datasets.

| Model | Variant | Quality Eval Acc (%) | Length Eval Acc (%) |
|---|---|---|---|
| Qwen2-1.5B-Base | LIFT-plus$_2^{reverse}$ | 58.78 | 85.58 |
| | LIFT-plus$_2^{noreverse}$ | 59.41 | 87.78 |
| | LIFT-plus$_2^{empty}$ | 58.33 | 89.71 |
| Qwen2-1.5B-Instruct | LIFT-plus$_2^{reverse}$ | 60.11 | 86.54 |
| | LIFT-plus$_2^{noreverse}$ | 63.17 | 85.21 |
| | LIFT-plus$_2^{empty}$ | 61.02 | 85.53 |
| Llama-3.1-8B-Base | LIFT-plus$_2^{reverse}$ | 51.34 | 91.32 |
| | LIFT-plus$_2^{noreverse}$ | 57.71 | 88.14 |
| | LIFT-plus$_2^{empty}$ | 50.27 | 88.78 |
| Llama-3.1-8B-Instruct | LIFT-plus$_2^{reverse}$ | 60.11 | 95.19 |
| | LIFT-plus$_2^{noreverse}$ | 56.65 | 92.31 |
| | LIFT-plus$_2^{empty}$ | 57.97 | 90.71 |

To further illustrate how reward models quickly overfit to length instructions during training, thereby impeding semantic learning, we visualize the accuracy trajectories of Qwen-2-1.5B-Instruct and Llama-3.1-8B-Instruct on $\mathcal{D}^q_{eval}$ and $\mathcal{D}^l_{eval}$ as training steps increase (see Figure 5). The results show that as training progresses, the model accuracy on $\mathcal{D}^l_{eval}$ increases rapidly, whereas the accuracy on $\mathcal{D}^q_{eval}$ improves at a much slower rate. Moreover, once the accuracy on $\mathcal{D}^l_{eval}$ reaches its peak, the accuracy on $\mathcal{D}^q_{eval}$ also begins to decline and gradually plateaus. This tendency strongly supports the conclusion that the reward models quickly prioritize learning length instructions, achieving high accuracy on length-related tasks, but at the expense of semantic quality. The slow improvement in the accuracy on $\mathcal{D}^q_{eval}$ further suggests that the reward models struggle to learn the necessary semantic understanding, as they are overly focused on conforming to length constraints.

## F ADDITIONAL EXPERIMENTAL RESULTS AND ANALYSIS

### F.1 THE EVALUATION DETAILS OF DPO MODELS

Similar to Appendix D.4, due to the length instruction $x_l$ in the AlpacaEval-LI benchmark provided by LIFT (Yuan et al., 2024) includes only the "or less" length instruction, it is necessary to incorporate the "or more" length instruction. Therefore, we extend the AlpacaEval-LI benchmark to include both the "or less" and "or more" length instructions. Specifically, in AlpacaEval-LI, $word\_num$ is set based on the shortest response length for the original prompt $x$ across three advanced models — GPT-4 (Achiam et al., 2023), Claude3-Opus, and Mistral Large — which we assume approximates the median response length for each prompt. This suggests that well-reasoned, detailed responses

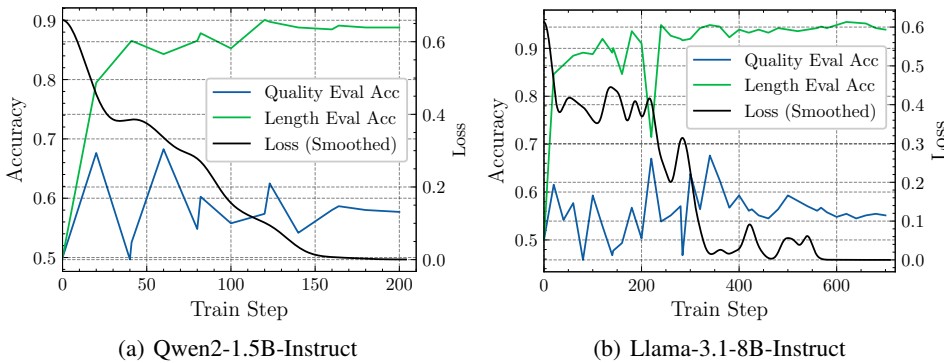

(a) Qwen2-1.5B-Instruct          (b) Llama-3.1-8B-Instruct

Figure 5: The trajectories of Quality Eval Acc, Length Eval Acc, and Training Loss (Smoothed) for different reward models trained on LIFT-plus$_2^{reverse}$ across training steps.

may exceed $word\_num$, while concise yet valid responses may fall below it. Therefore, we improve the AlpacaEval-LI benchmark by modifying the "or less" in $x_l$ to "or more", specifically replacing "Answer the following instruction using $\{word\_num\}$ or less" with "Answer the following instruction using $\{word\_num\}$ or more", while keeping all other aspects unchanged. This results in the expanded AlpacaEval-LI-plus-more benchmark, and the original AlpacaEval-LI benchmark is referred to as AlpacaEval-LI-plus-less benchmark. In cases where there is no ambiguity, we refer to AlpacaEval-LI-plus-less and AlpacaEval-LI-plus-more benchmark as AlpacaEval-LI-plus benchmark.

For the DPO evaluation metrics, as described in Section 5.1, we use automated assessment by leveraging GPT-4o (Hurst et al., 2024) to label each pair of responses as *win*, *tie*, or *lose* based on semantic quality. The prompt used to guide GPT-4o's evaluation is similar to the one provided in Appendix D.2, as shown in Figure 6. Specifically, we evaluate each DPO model's performance by comparing it with its corresponding SFT/*Instruct* model using three metrics: *Length Acc* refers to the accuracy of the model-generated response length that satisfies the length constraint specified in the given length instruction $x_l$ in AlpacaEval-LI-plus. *Length Win Ratio* denotes the proportion of cases in which the model-generated response is labeled as *win* compared to the response generated by the SFT/*Instruct* models, under the given length instruction $x_l$. *Quality Win Ratio* measures the proportion of cases in which the model-generated response is labeled as *win* compared to the response generated by the SFT/*Instruct* models under the given original prompt $x$. *Length Win Ratio* and *Quality Win Ratio* evaluate the semantic quality of responses generated by the DPO models under different prompts.

```
Response Semantic Quality Comparison Template

Given the following instruction and two candidate responses:

Instruction: {}
Response 1: {}
Response 2: {}

Please evaluate which response better follows the instruction based on the following criteria:
1. Adherence to the instruction.
2. Overall semantic quality.
3. Conciseness: the response should avoid redundancy, unnecessary verbosity, or inclusion of
   irrelevant information.

Based on your evaluation, respond in the format: 'Response 1 is better than Response 2', 'Response
2 is better than Response 1', or 'Response 1 is equal to Response 2'.
```

Figure 6: Response comparison template for evaluating the semantic quality between responses.

Table 10: *Length Eval Acc* of different reward models on length ($\mathcal{D}_{eval}^{l}$) evaluation datasets.

| Model | Baseline | LIFT-plus | Rc-RM |
|---|---|---|---|
| Qwen2-1.5B-Base | 52.41 | **87.18** | 86.22 |
| Qwen2-1.5B-Instruct | 51.77 | **86.54** | 84.61 |
| Qwen2.5-7B-Base | 52.24 | 84.94 | **88.14** |
| Qwen2.5-7B-Instruct | 52.88 | 83.97 | **92.31** |
| Llama-3.1-8B-Base | 51.92 | **91.32** | 88.46 |
| Llama-3.1-8B-Instruct | 49.04 | **95.19** | 93.27 |
| Gemma-2-9B-it | 51.28 | **91.66** | 90.91 |
| Qwen2.5-14B-Instruct | 54.13 | 95.25 | **95.34** |

## F.2 THE RESULTS OF REWARD MODELS

**Rc-RM effectively follows length instructions.** In addition to mitigating length bias, Rc-RM significantly enhances the ability to follow length instructions, as demonstrated by the *Length Eval Acc* results in Table 10. Specifically, Rc-RM achieves accuracy comparable to LIFT-plus and surpasses it in nearly half of the models, with gains of $3.2\%$ on Qwen2.5-7B-Base, $8.34\%$ on Qwen2.5-7B-Instruct, and $0.09\%$ on Qwen2.5-14B-Instruct.

Taking "or less" length instruction as a representative experiment, we evaluate the score consistency of reward models with respect to length instructions by constructing a sequence of length instructions $x_l^{(i,j)}, j = \{1, 2, ..., 8\}$, with increasing *word_num*. Specifically, we first select all instances $(x^{(i)}, y_c^{(i)}, y_r^{(i)})$ in $\mathcal{D}_{eval}$ where $|y_c^{(i)}| < |y_r^{(i)}|$, referring to this subset as $\mathcal{D}_{eval}^{less}$. For each $x^{(i)}$ in $\mathcal{D}_{eval}^{less}$, we then construct the following sequence of *word_num* values ($L_{wn}$) to generate the corresponding sequence of length instructions $x_l^{(i,j)}$, and refer to the entire constructed evaluation dataset as $\mathcal{D}_{eval}^{mls}$:

$$L_{wn} = [\, l_w - 2T, \, l_w - T, \, l_w, \, l_w + L, l_w + 2L, \, l_l, \, l_l + T, \, l_l + 2T \,],$$
$$\mathcal{D}_{eval}^{mls} = \{(x_l^{(i,j)}, y_w^{(i)}, y_l^{(i)})\}, \quad j \in \{1, 2, \dots, 8\}, \tag{22}$$

where $x_l^{(i,j)} = $ "Answer the following instruction using $\{word\_num = L_{wn}^{(j)}\}$ words or less." $+ x^{(i)}$, $T = 10$, $L = (l_l - l_w)/3$, $l_w = |y_c|$, and $l_l = |y_r|$. For the sequence of $(x_l^{(i,j)}, y_c^{(i)}, y_r^{(i)}), j = \{1, 2, ..., 8\}$ in $\mathcal{D}_{eval}^{mls}$, the ideal performance of the reward model should be as follows: Initially, when $word\_num < l_w$, the length constraint of $x_l^{(i,j)}$ is invalid for both $y_c^{(i)}$ and $y_r^{(i)}$, the difference in reward scores between $y_c^{(i)}$ and $y_r^{(i)}$ primarily reflect the semantic difference in the original data $(x^{(i)}, y_c^{(i)}, y_r^{(i)})$. When $word\_num = l_w$, the length constraint of $x_l^{(i,j)}$ aligns with $y_c^{(i)}$ but not $y_r^{(i)}$, resulting in a greater difference between $y_c^{(i)}$ and $y_r^{(i)}$ compared to their original semantic difference. When $word\_num = l_l$, the length constraint of $x_l^{(i,j)}$ becomes valid for both $y_c^{(i)}$ and $y_r^{(i)}$, and the difference reverts to those derived from the original semantics. Therefore, the difference in the predicted scores of the reward model for $y_c^{(i)}$ and $y_r^{(i)}$ should initially increase with the rise in $word\_num$, then decrease, eventually returning to the original semantic difference.

The results, as shown in Figure 3(c) and 3(d), demonstrate that both LIFT-plus and Rc-RM exhibit an initial increase followed by a decrease in reward score differences as $word\_num$ increases. However, LIFT-plus fails to return to the original difference after $word\_num$ increases, instead remaining at a level higher or lower than that observed when $word\_num < |y_c|$. This indicates that LIFT-plus overfits to the length instructions, thereby neglecting the original semantic instructions. In contrast, Rc-RM accurately reflects the expected behavior: the score difference increases with the rise in $word\_num$, then decreases, eventually returning to the original semantic difference. Moreover, Rc-RM exhibits minimal score fluctuation at both extremes – when length constraints are invalid for

both responses and when they are valid for both responses. This robust performance provides strong evidence for the effectiveness of our approach in simultaneously adhering to both length and original semantic instructions.

We conducted further experiments by training a policy using the trained reward models via PPO. We trained the *Qwen2-1.5B-Instruct* policy using the reward models of Qwen2-1.5B-Instruct and Qwen2.5-7B-Instruct from Table 4. The evaluation was carried out on AlpacaEval following the same metrics used for evaluating the DPO models. The results are in Table 11. It can be observed that the policy trained using Rc-RM exhibits high semantic quality. Its performance significantly surpasses other length-bias-mitigating methods, further demonstrating the effectiveness of our method.

Table 11: Evaluation results of different PPO models on AlpacaEval (Dubois et al., 2024b).

| Metrics | Qwen2-1.5B-Instruct | | |
|---|---|---|---|
| | Baseline | ODIN | Rc-RM |
| Quality Win Ratio (%) | 38.28 | 41.57 | **44.06** |
| Response Length | 671.49 | 342.85 | 308.16 |
| Metrics | Qwen2.5-7B-Instruct | | |
| | Baseline | ODIN | Rc-RM |
| Quality Win Ratio (%) | 40.31 | 45.03 | **49.26** |
| Response Length | 597.27 | 336.15 | 277.15 |

We further conducted experiments with the GRPO algorithm. In line with the PPO experiments presented above, we trained the *Qwen2-1.5B-Instruct* policy using the reward models of *Qwen2-1.5B-Instruct* and *Qwen2.5-7B-Instruct* from Table 4. The evaluation was again performed on AlpacaEval, reporting *Quality Win Ratio* as the primary metric. The results are summarized in Table 12. As shown, the findings are consistent with those obtained using PPO: policies trained with Rc-RM significantly outperform other length-bias-mitigating methods, further corroborating the effectiveness of our approach.

Table 12: Evaluation results of different GRPO models on AlpacaEval (Dubois et al., 2024b).

| Metrics | Qwen2-1.5B-Instruct | | |
|---|---|---|---|
| | Baseline | ODIN | Rc-RM |
| Quality Win Ratio (%) | 37.69 | 43.14 | **45.89** |
| Response Length | 719.37 | 498.22 | 330.15 |
| Metrics | Qwen2.5-7B-Instruct | | |
| | Baseline | ODIN | Rc-RM |
| Quality Win Ratio (%) | 41.23 | 47.25 | **51.60** |
| Response Length | 603.18 | 355.69 | 298.70 |

### F.3 THE RESULTS OF QWEN2.5-1.5B REWARD MODELS

This subsection mainly presents the RM experimental results for Qwen2.5-1.5B and some unreasonable experimental phenomena. As show in Table 13, for Qwen2.5-1.5B-Base, the results are similar to those to Section 5.2. Rc-RM not only achieves higher performance in *Quality Eval Acc*, being 7.69% higher than LIFT-plus and 6.41% higher than ODIN, but also performs almost equally with LIFT-plus in *Length Eval Acc*. For Qwen2.5-1.5B-Instruct, Rc-RM still outperforms LIFT-plus by 3.21% and ODIN by 1.61% in *Quality Eval Acc*. However, the results for LIFT-plus are somewhat unreasonable: its *Length Eval Acc* is only 67.63%, 12.5% lower than Rc-BT, while its *Quality Eval Acc* is 10.9% higher than Baseline.

Furthermore, we plot the *Quality Eval Acc* and *Length Eval Acc* trajectories for each method across training steps, as shown in Figure 7. It can be observed that in the early stages of training, LIFT-

Table 13: Evaluation results of reward models on quality ($\mathcal{D}^q_{eval}$) and length ($\mathcal{D}^l_{eval}$) evaluation datasets (Qwen2.5-1.5B).

| Metrics | Qwen2.5-1.5B-Base | | | | Qwen2.5-1.5B-Instruct | | | |
|---|---|---|---|---|---|---|---|---|
| | Baseline | LIFT-plus | ODIN | Rc-RM | Baseline | LIFT-plus | ODIN | Rc-RM |
| Quality Eval Acc (%) | 58.97 | 60.26 | 61.54 | **67.95** | 58.01 | 68.91 | 70.51 | **72.12** |
| Length Eval Acc (%) | 51.28 | **87.82** | 55.45 | 85.89 | 53.53 | 67.63 | 50.00 | **80.13** |

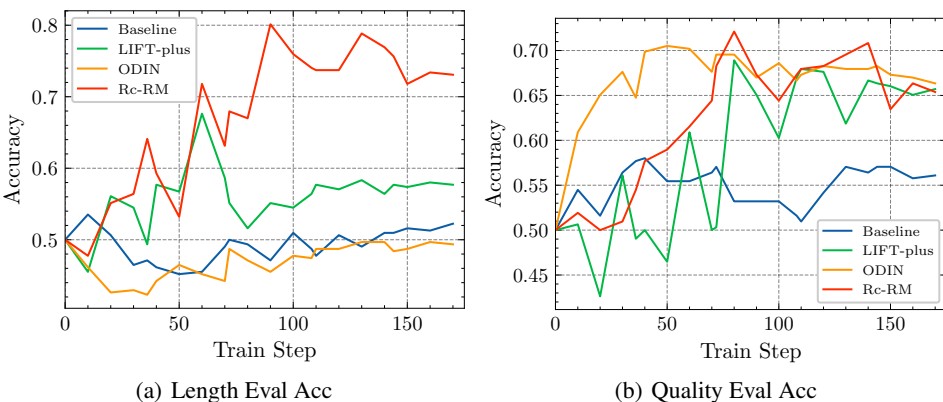

(a) Length Eval Acc  (b) Quality Eval Acc

Figure 7: The trajectories in Length Eval Acc and Quality Eval Acc of Qwen2.5-1.5B-Instruct with training steps.

plus's *Length Eval Acc* gradually increases, while its *Quality Eval Acc* is worse than that of Baseline, which aligns with our analysis in Section 3.4. However, after just 60 steps of training, LIFT-plus's *Length Eval Acc* suddenly drops drastically, while *Quality Eval Acc* begins to rise significantly. This ultimately results in *Length Eval Acc* stabilizing at a relatively low value and *Quality Eval Acc* stabilizing at a relatively high value. Since this is a single isolated example, we suspect it may be related to the specific model type and will conduct a more detailed analysis in future work.

## F.4 The Results of DPO Models in AlpacaEval

To further complement the results reported in Section 5.3, we conducted additional comparisons between our method and SamPO (Lu et al., 2024) by training on the same datasets used in Section 5. Specifically, we focused on two representative backbone models: Qwen2.5-7B-Instruct and Llama3.1-8B-Instruct. The results are evaluated on AlpacaEval, where we report *Quality Win Ratio* as the primary metric. The results are summarized in Table 14. models trained with Rc-DPO consistently maintain high semantic quality. More importantly, their performance substantially surpasses that of SamPO across both backbones, which further substantiates the robustness and effectiveness of our approach. These findings reinforce the conclusions drawn in Section 5.3, indicating that Rc-DPO not only mitigates length bias but also enhances overall response quality in alignment-oriented fine-tuning.

Table 14: Evaluation results of different DPO models on AlpacaEval.

| Metrics | Qwen2.5-7B-Instruct | | Llama-3.1-8B-Instruct | |
|---|---|---|---|---|
| | SamPO | Rc-DPO | SamPO | Rc-DPO |
| Quality Win Ratio (%) | 38.12 | **44.63** | 50.83 | **64.34** |

F.5 THE RESULTS OF DPO MODELS IN ALPACAEVAL-LI-PLUS-LESS

Due to length bias, models tend to generate longer responses, leading to higher *Length Acc* on the AlpacaEval-LI-plus-more benchmark, even when the models have no awareness of length instructions. This makes it difficult to fairly compare models' ability to follow length instructions. Therefore, by examining the performance on the AlpacaEval-LI-plus-less benchmark, we can better assess how effectively each method adheres to the specified length limits while still maintaining a high level of semantic quality in the generated responses.

The results on AlpacaEval-LI-plus-less are shown in Tabel 15. As observed, for the *Base* models, Baseline models are trained only on $\mathcal{D}_{sft}$ and $\mathcal{D}_{rm}$, and do not demonstrate the ability to follow length instructions. On the other hand, the *Instruct* models inherently exhibit a stronger ability to follow length instructions. Consequently, Baseline models trained on the *Instruct* models also acquire a certain degree of length instruction adherence. For R-DPO, since it was trained only on $\mathcal{D}_{rm}$, and according to the *Response Length* results on Tables 5, 15 and 17, it primarily reduces the response length across all instructions (include "or more" length instruction), without explicitly following the length constraints. This indicates that R-DPO does not specifically address length instruction adherence, but simply focuses on minimizing the response length overall. Thus, the main comparison method we focus on here is LIFT-plus.

Table 15: Evaluation results of different DPO models on AlpacaEval-LI-plus-less.

| Metrics | Qwen2.5-7B-Base | | | | Qwen2.5-7B-Instruct | | | |
|---|---|---|---|---|---|---|---|---|
| | Baseline | LIFT-plus | R-DPO | Rc-DPO | Baseline | LIFT-plus | R-DPO | Rc-DPO |
| Length Acc (%) | 5.74 | **86.78** | 8.35 | 82.04 | 89.65 | **100** | 90.90 | **100** |
| Response Length | 544.47 | 106.87 | 637.44 | 177.40 | 136.75 | 23.56 | 131.17 | 108.24 |
| Length Win Ratio (%) | 34.04 | 1.37 | 34.54 | **44.14** | 32.17 | 2.99 | 31.92 | **50.75** |

| Metrics | Llama-3.1-8B-Base | | | | Llama-3.1-8B-Instruct | | | |
|---|---|---|---|---|---|---|---|---|
| | Baseline | LIFT-plus | R-DPO | Rc-DPO | Baseline | LIFT-plus | R-DPO | Rc-DPO |
| Length Acc (%) | 13.59 | **98.75** | 27.68 | 94.51 | 87.66 | **100** | 87.03 | **100** |
| Response Length | 424.47 | 6.62 | 518.74 | 118.50 | 150.43 | 35.01 | 151.39 | 89.75 |
| Length Win Ratio (%) | 35.41 | 0.25 | 47.88 | **64.96** | 43.89 | 1.00 | 43.77 | **64.71** |

It is evident that LIFT-plus suffers from a severe "short bias" issue in Appendix F.8, focusing exclusively on the "or less" length instruction $x_l$ itself without considering the specified *word_num* or the original prompt $x$. This leads to extremely short responses that strictly comply with the length instruction but completely disregard the semantic requirement of the original prompt. Specifically, On Qwen2.5-7B-Base and Qwen2.5-7B-Instruct, the average response lengths generated by LIFT-plus are only 106.87 and 23.56, significantly below the average length constraint of 180.23 in AlpacaEval-LI-plus-less. On Llama-3.1-8B-Instruct, the average response length is 35.01, and on Llama-3.1-8B-Base, the average response length drops to a mere 6.62, showing a 111.88-word difference compared to Rc-BT. This behavior results in LIFT-plus achieving nearly $100\%$ *Length Acc*, but with *Length Win Ratio* close to $0\%$, which fails to align with the desired semantic quality.

In contrast, Rc-DPO not only adheres to the "or less" length instruction but also considers the specified *word_num* and original prompt. It strives to maximize the semantic quality of the response within the length constraint. Therefore, compared to the *Quality Win Ratio* in Table 5, Rc-DPO maintains semantic quality even under length constraints. Notably, on Qwen2.5-7B-Base and Llama-3.1-8B-Base, Rc-BT achieves $82.04\%$ and $94.51\%$ *Length Acc*, respectively, while boosting the *Length Win Ratio* to $44.14\%$ and $64.96\%$, further enhancing the semantic abilities of SFT models. For both Qwen2.5-7B-Instruct and Llama-3.1-8B-Instruct, Rc-BT achieves $100\%$ *Length Accuracy*, while obtaining *Length Win Ratio* of $50.75\%$ and $64.71\%$, respectively. This fully demonstrates the effectiveness of Rc-DPO in balancing adherence to length instructions with high semantic quality.

To provide a more detailed analysis of how well our DPO models follow length instructions, we further evaluate on AlpacaEval-LI-plus-less. For each length instruction $x_l$ in AlpacaEval-LI-plus-less, we adjust the original length constraint by $\pm50$ and $\pm100$ to create five benchmarks, denoted $A_{-100}$, $A_{-50}$, $A_0$, $A_{50}$, $A_{100}$. These benchmarks allow us to assess each model's ability to dynamically ad-

just response length. The results are shown in the Table 16 (with the average length constraint of each benchmark indicated in parentheses).

Since some original length constraints are below 100, the corresponding values in $A_{-100}$ and even $A_{-50}$ would become negative. We therefore remove these samples from $A_{-100}$ and $A_{-50}$, which results in the average length constraint of $A_{-50}$ ($A_{-100}$) not being exactly 50 (100) less than that of $A$. As the baseline and R-DPO models perform poorly in following length instructions (see Table 15), let alone in dynamic adherence, we primarily compare against LIFT-plus. As the table shows, LIFT-plus fails to dynamically follow varying constraints and merely shortens responses indiscriminately. In contrast, our method effectively adjusts response length in accordance with the changing length instructions, clearly demonstrating its effectiveness.

Table 16: Evaluation results of different DPO models on AlpacaEval-LI-plus-less with different length constraints.

| Metrics | Qwen2.5-7B-Instruct | | Llama3.1-8B-Instruct | |
|---|---|---|---|---|
| | LIFT-plus | Rc-DPO | LIFT-plus | Rc-DPO |
| $A_{-100}$ (124.27) | 23.01 | **78.29** | 30.14 | **65.73** |
| $A_{-50}$ (149.42) | 24.71 | **85.17** | 36.83 | **72.69** |
| $A$ (180.24) | 23.56 | **108.24** | 35.01 | **89.75** |
| $A_{50}$ (230.24) | 25.78 | **130.50** | 37.19 | **121.14** |
| $A_{100}$ (280.24) | 26.14 | **164.07** | 42.93 | **159.88** |

### F.6 THE RESULTS OF DPO MODELS IN ALPACAEVAL-LI-PLUS-MORE

In this subsection, we present the experimental results of the DPO models from Section 5.3 on AlpacaEval-LI-plus-more, as shown in Table 17. Similar to Section 5.3, since R-DPO was not trained on length instruction data, we primarily compare our method with LIFT-plus. First, as seen in Table 5, due to the influence of length bias, Baseline model outputs longer responses, which means that it largely satisfies the "or more" length instruction limit without focusing on the length constraint itself. Moreover, a comparison with Table 17 further confirms it: after adding the "or more" length instruction limit, the length variation in the Baseline responses is not significant (especially for the *Instruct* model). At the same time, its *Length Win Ratio* demonstrates that compared to the SFT / *Instruct* models, the Baseline's responses are more redundant and repetitive, rather than truly improving semantic quality, leading to no significant improvement in *Length Win Ratio*. In fact, for all four models, the Baseline's *Length Win Ratio* is lower than ours by $17.46\%$, $24.31\%$, $9.61\%$, and $14.71\%$, respectively.

For LIFT-plus, the results are similar to those in Appendix F.8, where it focuses only on the "or more" length instruction and disregards the semantic requirements of the original prompt and the specific length constraints of the "or more" length instruction. This causes the model to generate excessively long and unnecessary responses, resulting in a high *Length Acc* but a decline in *Length Win Ratio*. Specifically, for the *Instruct* models, due to the strong foundational instruct-following capability, the over-generation behavior of LIFT-plus is relatively mild, with response lengths generally controlled within $200 \sim 400$. Its *Length Win Ratio* is only lower than Rc-DPO by $33.62\%$ and $26.18\%$. However, for the *Base* models, LIFT-plus's response length skyrockets above 600, causing severe semantic redundancy, and its *Length Win Ratio* significantly drops, being lower than Rc-DPO by $39.53\%$ and $35.91\%$. This result, when analyzed alongside the findings in Section 5.3 and Appendix F.8, fully illustrates the drawbacks of LIFT-plus's over-reliance on length instructions and the complementary nature of Rc-BT in terms of both length instruction and semantic quality.

### F.7 ABLATION STUDIES AND EXTRA RESULTS

Due to the high cost of evaluating the DPO model, we primarily use the reward model for ablation studies. First, we conduct ablation studies to validate the key design choices of our approach. For convenience, we denote $\{(x, x_l^1, y_c)\}$ as $\mathcal{D}_{Rc}^c$ and $\{(x_l^2, x, y_r)\}$ as $\mathcal{D}_{Rc}^r$.

Table 17: Evaluation results of different DPO models on AlpacaEval-LI-plus-more.

| Metrics | Qwen2.5-7B-Base | | | | Qwen2.5-7B-Instruct | | | |
|---|---|---|---|---|---|---|---|---|
| | Baseline | LIFT-plus | R-DPO | Rc-DPO | Baseline | LIFT-plus | R-DPO | Rc-DPO |
| Length Acc (%) | 99.75 | **99.88** | 99.38 | **99.88** | 84.29 | 99.88 | 78.43 | **100** |
| Response Length | 680.84 | 550.27 | 682.66 | 319.05 | 249.99 | 303.16 | 238.03 | 263.30 |
| Length Win Ratio (%) | 27.93 | 5.86 | 35.16 | **45.39** | 35.79 | 26.48 | 26.93 | **60.10** |
| Metrics | Llama-3.1-8B-Base | | | | Llama-3.1-8B-Instruct | | | |
| | Baseline | LIFT-plus | R-DPO | Rc-DPO | Baseline | LIFT-plus | R-DPO | Rc-DPO |
| Length Acc (%) | 89.03 | **99.75** | 93.39 | 97.51 | 98.25 | **99.88** | 97.01 | **99.88** |
| Response Length | 448.77 | 664.20 | 540.34 | 429.69 | 279.04 | 413.42 | 246.73 | 279.56 |
| Length Win Ratio (%) | 36.03 | 9.73 | **47.76** | 45.64 | 45.14 | 33.67 | 47.01 | **59.85** |

$\mathcal{D}_{Rc}^c$ **and** $\mathcal{D}_{Rc}^r$ **are complementary.** We begin by proving the necessity of $\mathcal{D}_{Rc}^c$ and $\mathcal{D}_{Rc}^r$. To this end, we conduct ablation experiments using $\mathcal{D}_{rm} \cup \mathcal{D}_{Rc}^c$ (*w/o* $\mathcal{D}_{Rc}^r$) and $\mathcal{D}_{rm} \cup \mathcal{D}_{Rc}^r$ (*w/o* $\mathcal{D}_{Rc}^c$), with the results shown in Table 6. As observed, when only one type of augmented dataset is used, the RM's *Quality Eval Acc* drops significantly, approaching the Baseline results in Table 4. Simultaneously, its *Length Eval Acc* hovers around $50\%$, indicating that it fails to learn the length instruction. To further investigate, we prepend length instructions to each $x$ in $\mathcal{D}_{eval}^q$, forming $x_l$, where $y_c$ satisfies the length constraint in the instruction, , resulting in $\mathcal{D}_{eval}^{q,l}$. We then use the DPO models in Table 6 to score the pairs $(x_l, y_c)$ in $\mathcal{D}_{eval}^{q,l}$ and $(x, y_c)$ in $\mathcal{D}_{eval}^q$ to observe changes in the RM's predicted scores. In theory, for the pair $(x_l, y_c)$, since $y_c$ satisfies both the length constraint and original prompt of $x_l$, it should outperform $(x, y_c)$. That is, the score of reward model for $(x_l, y_c)$ should be higher than that for $(x, y_c)$. However, the results, illustrated in Figure 8(a), reveal that for the RM trained on $\mathcal{D} \cup \mathcal{D}_c$, the scores of $(x_l, y_c)$ with added length instructions are consistently lower than those of the original $(x, y_c)$, regardless of whether the added length constraint matches $y_c$'s length. This indicates that after training on $\mathcal{D}_{rm} \cup \mathcal{D}_{Rc}^c$, the RM learns a *length instruction bias*. Specifically, in $\mathcal{D}_{rm} \cup \mathcal{D}_{Rc}^c$, for any $(x_l, y_c)$, with an added length instruction, the reward model merely reduces the score compared to the original $(x, y_c)$ pair, without paying attention to the length instruction itself. A similar phenomenon in Figture 8 is observed in the RM trained on $\mathcal{D}_{rm} \cup \mathcal{D}_{Rc}^r$. Therefore, the combination of $\mathcal{D}_{Rc}^c$ and $\mathcal{D}_{Rc}^r$ is necessary to prevent the reward model from being exploited by this bias hacking.

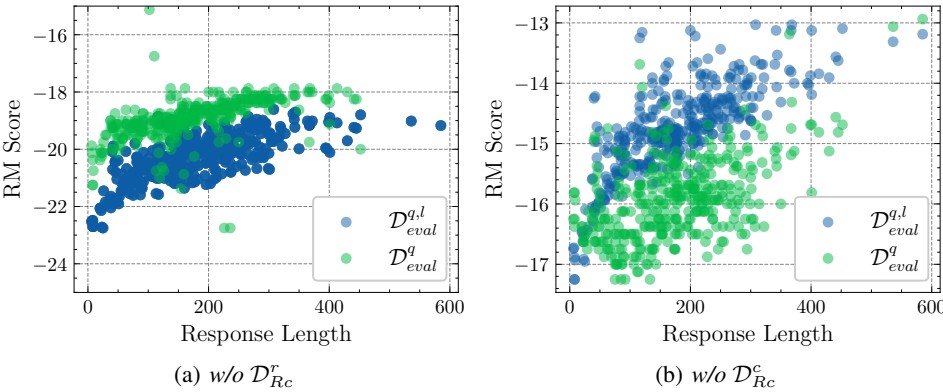

(a) *w/o* $\mathcal{D}_{Rc}^r$         (b) *w/o* $\mathcal{D}_{Rc}^c$

Figure 8: The relationship between response length and RM score evaluated on $\mathcal{D}_{eval}^q$ and $\mathcal{D}_{eval}^{q,l}$ for Rc-RM trained with Llama-3.1-8B-Instruct.

**Effectiveness of different ratios of** $\mathcal{D}_{Rc}$**.** We conducted further experiments to demonstrate the efficiency of the augmented dataset $\mathcal{D}_{Rc}$ we constructed. Specifically, we gradually increased $\mathcal{D}_{Rc}$ from $0\%$ to $100\%$ on the original training dataset $\mathcal{D}_{rm}$, with a $10\%$ increment, to train the reward models for *Qwen2-1.5B-Instruct* and *Qwen2.5-7B-Instruct*. The results of *Quality Eval Acc* (%) are shown in Table 18. It can be observed that our method demonstrates superlinear performance

improvements with a smaller proportion of $\mathcal{D}_{Rc}$. In other words, using only $40\%$ of the augmented data yields approximately a $70\%$ performance improvement. This indicates that, even under tight computational constraints, our method can achieve significant performance gains with the addition of a small amount of data, which fully demonstrates the high efficiency of our augmented dataset $\mathcal{D}_{Rc}$ and our method.

Table 18: *Quality Eval Acc* of different reward models on quality ($\mathcal{D}_{eval}^{q}$) evaluation datasets under different proportions of $\mathcal{D}_{Rc}$.

| Model | 0% | 10% | 20% | 30% | 40% | 50% | 60% | 70% | 80% | 90% | 100% |
|---|---|---|---|---|---|---|---|---|---|---|---|
| Qwen2-1.5B-Instruct | 60.75 | 64.43 | 65.69 | 66.68 | 67.31 | 68.11 | 68.77 | 69.95 | 70.40 | 71.05 | 71.47 |
| Qwen2.5-7B-Instruct | 59.31 | 61.92 | 64.26 | 66.33 | 68.12 | 69.63 | 70.87 | 71.83 | 72.52 | 72.93 | 73.07 |

**Effectiveness of different ratios of $\mathcal{D}_{Rc}^{c}$ and $\mathcal{D}_{Rc}^{r}$ mixtures.** We also briefly explore the impact of different ratios of $\mathcal{D}_{Rc}^{c}$ and $\mathcal{D}_{Rc}^{r}$ on the final RM results. Specifically, we fix $\mathcal{D}_{rm}$ and $\mathcal{D}_{Rc}^{c}$ constant, and increase $\mathcal{D}_{Rc}^{r}$ from $0\%$ to $100\%$ with a $10\%$ increment. The results are shown in Figure 9 (a) and (b). As observed, initially, the model is biased toward the one-sided data of $\mathcal{D}_{Rc}^{c}$, leading to low *Quality Eval Acc* and *Length Eval Acc*, even falling below the Baseline models (denoted as green dashed line). As $\mathcal{D}_{Rc}^{r}$ increased, *Length Eval Acc* rises rapidly, with the growth rate slowing down around $30\%$, while *Quality Eval Acc* gradually increases. It effectively demonstrates the complementary effect of $\mathcal{D}_{Rc}^{c}$ and $\mathcal{D}_{Rc}^{r}$ in mitigating length bias and enhancing adherence to length instructions.

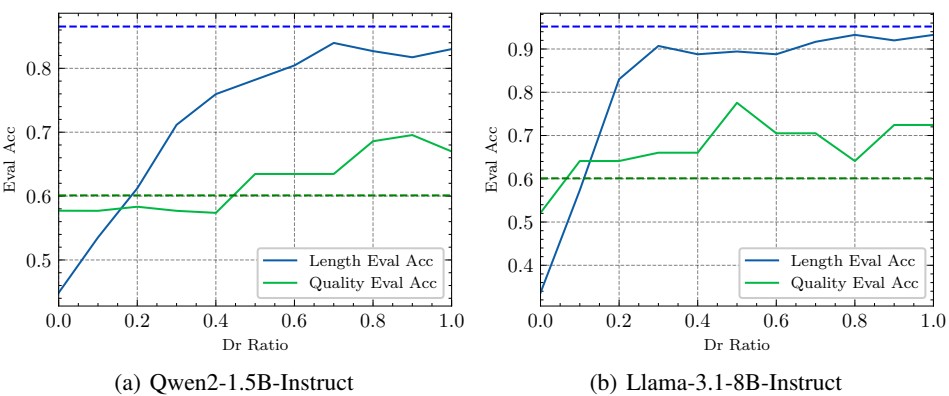

(a) Qwen2-1.5B-Instruct          (b) Llama-3.1-8B-Instruct

Figure 9: Variations in Rc-RM's *Length Eval Acc* and *Quality Eval Acc* as $\mathcal{D}_{Rc}^{r}$ increases. The green dashed line denotes the Baseline *Quality Eval Acc*, while the blue dashed line represents the *Length Eval Acc* of LIFT-plus.

**Effectiveness of $\mathcal{L}_{Rc}$.** We further conducted experiments by training the reward model using only the Rc-BT loss $\mathcal{L}_{Rc}$, emoving the standard BT loss while keeping all other training settings identical to Section 5. The evaluation was performed on the quality evaluation set $\mathbf{D}_{eval}^{f}$ (*Quality Eval Acc*) and the results are reported in Table 19. As shown, even without the BT loss, our method still outperforms the comparison methods, indicating that Rc-BT alone is sufficient for learning semantic preferences. These results further underscore the complementary nature of Rc-BT and BT, as combining them yields more consistent improvements in both semantic quality and length adherence.

**Impact of dataset size normalization.** Since our method constructs two examples from each training pair, $(x, y_c)$ and $(x, y_r)$, it is essentially equivalent to duplicating the original dataset rather than enlarging it with new data. To ensure fairness, we instead upsampled the baseline dataset to match our method's size and conducted reward model experiments on *Qwen2.5-1.5B-Instruct* and *Qwen2.5-7B-Instruct*. As shown in Table 20, our method still achieves higher *Quality Eval Acc*, significantly outperforming both the Baseline and other length-bias mitigation methods.

Table 19: Evaluation results of Rc-RM *only* $\mathcal{L}_{Rc}$ on quality ($\mathcal{D}^q_{eval}$) evaluation datasets.

| Metrics | Qwen2.5-1.5B-Instruct | | Qwen2.5-7B-Instruct | |
|---|---|---|---|---|
| | Rc-RM (*w/o* BT) | Rc-RM | Rc-RM (*w/o* BT) | Rc-RM |
| Quality Eval Acc (%) | 66.13 | **71.47** | 69.89 | **73.07** |

Table 20: Evaluation results of Rc-RM *same dataset size* $\mathcal{L}_{Rc}$ on quality ($\mathcal{D}^q_{eval}$) evaluation datasets.

| Metrics | Qwen2.5-1.5B-Instruct | | | Qwen2.5-7B-Instruct | | |
|---|---|---|---|---|---|---|
| | Baseline | ODIN | Rc-RM | Baseline | ODIN | Rc-RM |
| Quality Eval Acc (%) | 61.02 | 63.92 | **71.47** | 61.27 | 68.19 | **73.07** |

**Effectiveness across different datasets.** To further validate the generalizability of Rc-BT, we conduct RM training on the HH-RLHF (Bai et al., 2022a) dataset[9], while still using $\mathcal{D}^q_{eval}$ and $\mathcal{D}^l_{eval}$ for evaluation. The results are shown in Table 21. As seen, despite the HH-RLHF dataset being from a different domain compared to the OpenAssistant dataset (Köpf et al., 2024), which can be considered an out-of-distribution (OOD) scenario, Rc-RM still significantly outperforms other methods on $\mathcal{D}^q_{eval}$, while achieving results similar to LIFT-plus on $\mathcal{D}^l_{eval}$. Specifically, for Qwen2-1.5B-Instruct, Rc-RM outperforms Baseline by 2.57% and ODIN by 1.60%. For Qwen2.5-7B-Instruct, Rc-RM surpasses Baseline by 2.88% and ODIN by 0.32%. Notably, for Llama-3.1-8B-Instruct, Rc-RM significantly outperforms Baseline by 6.09% and also surpasses ODIN by 5.13%, further demonstrating the effectiveness of Rc-RM in mitigating length bias. Moreover. on $\mathcal{D}^l_{eval}$, Rc-RM achieves results comparable to LIFT, reaffirming its effectiveness in following length instructions. Finally, the consistent performance across different datasets provides strong evidence of the robustness and effectiveness of Rc-BT in diverse scenarios.

Table 21: Evaluation results of different reward models on quality ($\mathcal{D}^q_{eval}$) and length ($\mathcal{D}^l_{eval}$) evaluation datasets (HH-RLHF).

| Model | Variant | Quality Eval Acc (%) | Length Eval Acc (%) |
|---|---|---|---|
| Qwen2-1.5B-Instruct | Baseline | 58.65 | 59.36 |
| | LIFT-plus | 58.01 | **85.90** |
| | ODIN | 59.62 | 48.72 |
| | Rc-RM | **61.22** | 85.26 |
| Qwen2.5-7B-Instruct | Baseline | 62.50 | 52.24 |
| | LIFT-plus | 57.37 | **92.63** |
| | ODIN | 65.06 | 57.69 |
| | Rc-RM | **65.38** | 91.03 |
| Llama-3.1-8B-Instruct | Baseline | 63.14 | 51.60 |
| | LIFT-plus | 58.33 | **93.27** |
| | ODIN | 64.10 | 50.00 |
| | Rc-RM | **69.23** | 91.03 |

**Effectiveness across different bias.** We used format bias as another test bed, as it is the another common type of bias observed in LLMs. We used the FormatBiasEval dataset (Long et al., 2024), which consists of multiple-choice questions with specific formatting requirements, to train and evaluate reward models. As the dataset comes with a predefined split into training and evaluation subsets, we adhere to this split in our experiments. To adapt the training subset into a preference dataset, we designated the correct option as the chosen response and randomly selected incorrect option as

---

[9]We use *Instruct* models for direct RM training on the HH-RLHF dataset.

the rejected response. Then we constructed additional "response-conditioned" triples of the form $(x, x_f^1, y_c)$ and $(x, x_f^2, y_c)$, where $y_c$ violates the format constraints in $x_f^1$ (or $x_f^2$) while $y_r$ satisfies it. The final augmented dataset is defined as $\mathcal{D}_{rm}^f = \{(x, x_f^1, y_c)\} \cup \{(x, x_f^2, y_c)\}$.

Due to the programmatically constructible nature of the format, we created a semantic evaluation dataset based on the evaluation subset, referred to as $\mathcal{D}_{eval}^f$, using a methodology similar to that used for constructing $\mathcal{D}_{eval}^q$, but without relying on GPT-4o. The process is as follows: For each triple $(x, y_c, y_r)$ in the original evaluation dataset, we generated two complementary triples: $(x, f(y_c), y_r)$ and $(x, y_c, f(y_r))$, where $f(y)$ denotes a program that applies a randomly selected format pattern to the response $y$. Therefore, the $D_{eval}^{q,f}$ can effectively mitigate the impact of format bias and facilitate more fair evaluation of the reward model's semantic capabilities.

Table 22: The accuracy of different reward models on quality ($\mathcal{D}_{eval}^f$) evaluation datasets.

| Model | Baseline | Pc-Format | Rc-RM |
|---|---|---|---|
| Qwen2-1.5B-Instruct | 76.25 | 75.41 | **79.17** |
| Qwen2.5-7B-Instruct | 82.42 | 78.33 | **89.27** |
| Llama-3.1-8B-Instruct | 79.39 | 79.01 | **84.24** |

We conduct experiments on *Instruct* models. And the final results are shown in Table 22, where we refer to the prompt-conditioned method, which is similar to LIFT, as **Pc-Format**. As shown, Rc-RM remains more effective in mitigating format bias, achieving superior accuracy across all evaluated models. These results clearly highlight the strong generalization ability of our method.

**Effectiveness of eneration + verification pipeline.** We further employed another strong LLM, Claude-Sonnet-4.5, to rewrite the triplets $(x_l^{(i)}, y_c^{(i)}, y_r^{(i)})$ using the same pipeline described in Appendix D.2, generating a new evaluation dataset $\mathcal{D}_{eval}^{q'}$. We then evaluated all methods on this dataset. The results (*Quality Eval Acc*) are as shown in Table 23. The results demonstrate that the performance trends across different models remain consistent even when evaluated on datasets rewritten by different LLMs, further validating the robustness of our evaluation data construction pipeline.

Table 23: *Quality Eval Acc* of different reward models on quality ($\mathcal{D}_{eval}^{q'}$) evaluation datasets.

| Model | Baseline | LIFT | ODIN | Rc-RM |
|---|---|---|---|---|
| Qwen2.5-7B-Instruct | 61.74 | 57.32 | 68.15 | **74.89** |
| Llama3.1-8B-Instruct | 58.13 | 58.67 | 64.34 | **72.21** |

Additionally, we incorporated two types of automatic metrics: BERTScore (BERTScore-F1) to quantify compression-retention and n-gram Repetition Rate ($n = 3$) to measure redundancy. Specifically, For each generation iteration, we computed these metrics on the rewritten responses. The results are shown in Table 24. Thanks to the strong capability of GPT-4o and our generation-verification pipeline, the initial BERTScores were already high while redundancy remained low. Further, low-quality candidates were consistently filtered out through iterative rejection, ensuring that the final responses were both semantically faithful and non-redundant.

Table 24: BERTScores and n-gram Repetition Rate of different turn of rewriting.

| Model | First Rewrite | Second Rewrite | Third Rewrite |
|---|---|---|---|
| BERTScore | 0.912 | 0.974 | 0.981 |
| n-gram Repetition Rate | 0.27% | 0.09% | 0.06% |

We also include qualitative examples of discarded failure cases (Figure 10) in earlier rewriting rounds for reference. These results collectively demonstrate that, despite the potential risk of semantic or stylistic drift in LLM rewrites, our iterative generation pipeline effectively identifies and

filters out low-quality responses, ultimately ensuring that the final outputs exhibit strong semantic retention and minimal redundancy.

---

**query:** What types of tests do we have in software development?

**original response:** Software testing is generally classified into two main broad categories: functional testing and non-functional testing. There is also another general type of testing called maintenance testing.\n\nFunctional testing involves the testing of the functional aspects of a software application. When you're performing functional tests, you have to test each and every functionality. You need to see whether you're getting the desired results or not.\n\nThere are several types of functional testing, such as:\n\nUnit testing\nIntegration testing\nEnd-to-end testing\nSanity testing\nRegression testing\nAcceptance testing\nInterface testing\n\nNon-functional testing is the testing of non-functional aspects of an application, such as performance, reliability, usability, security, and so on. Non-functional tests are performed after the functional tests.\n\nNon-functional tests are not generally run manually. In fact, it's difficult to perform this kind of tests manually. So these tests are usually executed using tools.\n\nThere are several types of non-functional testing, such as:\n\nPerformance testing\nSecurity testing\nLoad testing\nFailover testing\nCompatibility testing\nUsability testing\n\nBesides performing tests, measuring the effectiveness of the tests is also important, and test coverage tells the effectiveness of your tests.\n\nYou can use programs called linters to check if the code of your software project meets the specified coding convention rules. A linter actually saves you a lot of time as manually checking the code written by several developers is a very time-consuming process.

**first rewrite (discarded due to semantic loss):** Non-functional testing focuses on aspects like performance, reliability, security, usability, and compatibility. These tests are usually automated and include performance, load, failover, and similar evaluations.

**query:** What's a good prompt to generate photorealistic objects in Stable Diffusion?

**original response:** There are several versions of Stable Diffusion which each respond differently to prompts. The best prompt is also likely to vary depending on the object in question and other context. For these reasons it would be difficult for me to provide an exact prompt. In general I would recommend including terms like \"photorealistic\", \"ultrarealistic\", \"photograph\", or \"HD camera\".

**first rewrite (discarded due to redundancy):** There are many different versions and checkpoints of Stable Diffusion, and each one can react quite differently to the same prompt. Because of this variation, as well as the fact that the ideal prompt depends heavily on the specific object you want to generate and the broader visual context you have in mind, it's difficult to give a single, universally "best" prompt. In general, though, if your goal is to create photorealistic objects, it usually helps to include descriptive keywords such as "photorealistic," "ultrarealistic," "high-detail photograph," "real-world lighting," or "shot with an HD camera." These kinds of terms tend to encourage the model to generate images with a more lifelike appearance and stronger photographic qualities.

---

Figure 10: Failure cases identified by the generation + verification pipeline.

## F.8 THE "SHORT BIAS" IN LIFT-PLUS

From the perspective of *Response Length* in Table 5, it might appear that LIFT-plus exhibits smaller length bias. This is not the case. Our results demonstrate that while LIFT-plus reduces response length, it does so at the expense of disregarding the balance between response length and semantic quality, resulting in suboptimal performance in both semantic quality and adherence to length instructions. Specifically, our results indicate that DPO models trained with the LIFT-plus method tend to exhibit a "short bias" phenomenon, where the model prioritizes generating shorter responses without considering semantic quality, given the "or less" instruction $x_l$ or the original prompt $x$. As shown in Table 5 and 15, both for *Quality Win Ratio* and *Length Win Ratio*, LIFT-plus significantly underperforms compared to R-DPO and Rc-BT. Notably, for Qwen2.5-7B-Base and Qwen2.5-7B-Instruct, LIFT-plus's *Quality Win Ratio* is lower than R-DPO by $8.73\%$ and $8.47\%$, and lower than Rc-BT by $13.72\%$ and $18.94\%$, respectively. A similar trend is observed with Llama-3.1-8B-Base and Llama-3.1-8B-Instruct. Furthermore, we plot the distribution of response lengths generated by LIFT-plus trained on Llama-3.1-8B-Instruct under the length instruction $x_l$ in AlpacaEval-LI-plus-less benchmark, as shown in Figure 11. As observed, the mean response length specified by the length instruction $x_l$ in AlpacaEval-LI-plus-less is $180.23$, while the response lengths generated by LIFT-plus are all below $100$. Furthermore, more than $90\%$ of the responses have length under $64$. This clearly demonstrates that the DPO models trained with LIFT-plus focus solely on the "or less"

length instruction $x_l$, disregarding the semantic requirements of the original prompt $x$ and even overlooking the specific length constraint (*word_num*) of the $x_l$ itself, resulting in the "short bias".

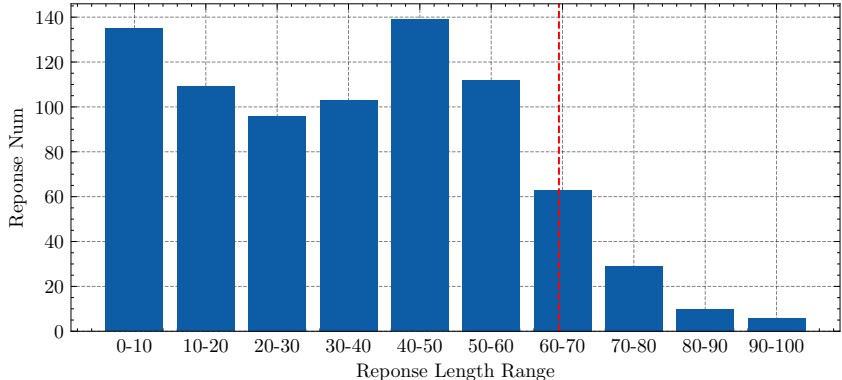

Figure 11: Response length range generated by LIFT-plus trained on Llama-3.1-8B-Instruct using length instruction $x_l$ in AlpacaEval-LI-plus-less benchmark. The red dashed line $x = 64$ represents the $90\%$ threshold, indicating that $90\%$ of the responses have a length less than or equal to $64$.

## G LLM USAGE

We made limited use of the LLM exclusively for minor linguistic refinements. Specifically, the LLM was used to polish the wording of a small number of sentences and phrases to improve clarity and readability. The LLM did not contribute to research ideation, experimental design, analysis, or the substantive writing of the paper. All conceptual, methodological, and technical contributions are entirely the work of the authors.

