# OpenReview forum: "Disentangling Length Bias in Preference Learning via Response-Conditioned Modeling"
_ICLR.cc/2026/Conference — ICLR 2026 Poster_

### Official Review · Reviewer_tJtt · 2025-10-25

**Soundness:** 3
**Presentation:** 2
**Contribution:** 2
**Rating:** 4
**Confidence:** 3

**Summary:**

This work tackles length bias in DPO. It first uses four focused experiments to show that (i) length bias genuinely exists, (ii) the evaluation set itself carries length bias, (iii) an implicit preference for longer answers ≠ the ability to follow explicit length instructions, and (iv) explicitly teaching length is easy but harms semantic quality. The experimental designs are insightful. Building on these findings, the authors propose a method that converts “implicit length bias” into an explicit understanding of length and disentangles it from semantic preference. They apply this idea to both reward modeling (RM) and DPO, and demonstrate effectiveness and generalization across multiple base models.

**Strengths:**

1. Clear motivation and an important problem. The paper targets a widely observed and practically consequential issue in RLHF/DPO—length bias. The authors establish motivation via a “falsify first, then model” storyline, using evidence to demonstrate the problem before proposing a solution.

2. The four small studies form a coherent chain: (existence) → (evaluation bias) → (implicit ≠ explicit) → (explicit hurts semantics). This progression is logically tight and highly readable, making the diagnosis convincing and the need for a new method evident.

3.  The core idea that explicitly modeling and disentangling a spurious format constraint from semantic preference appears portable to other instruction/format biases (e.g., politeness, number of bullet points, templated structure). This suggests a promising research trajectory beyond length alone.

**Weaknesses:**

1. 3.3 (length-compliance eval) construct validity: The eval set is built from model-rewritten pairs assumed to be semantically equivalent. Any residual semantic or stylistic drift can confound the claim “implicit length bias ≠ explicit length control.” Add a simple length-only heuristic baseline (choose the candidate that satisfies the numeric constraint) as a sanity upper bound, and include human spot checks for equivalence.

2. 3.4 (explicitly teaching length)  might result in overfitting vs. distribution shift: The {x_l, y_w, y_l} format makes length the most salient signal. The quality drop might reflect not only “overfitting to length,” but also a mismatch between synthetic length-instruction training data and the de-biased quality eval.

3. RC-DPO is essentially in the DPO family. Comparing mainly to ODIN (more of an RM-regularization baseline) is insufficient. Please add strong DPO variants such as SIMPO, CPO, etc, and evaluate on standard leaderboards.

4. Both the de-biased quality set and the length set depend on strong-LLM rewrites, risking style/domain shift and label noise. Strengthen with: (i) a human agreement study on a sampled subset; (ii) cross-source replication (another rewrite model or a small human-curated slice); and (iii) reporting failure cases where “meeting the numeric length” reduces information density (add redundancy/compression-retention metrics).

**Questions:**

See weakness.

---

> ### Author Response · Authors · 2025-11-19
>
> Thank you for your thoughtful and constructive feedback. We have carefully revised the manuscript to address your comments and improve its clarity and rigor. In the following, we respond to each of your points individually, with **all corresponding changes clearly marked in $\textcolor{green}{\textbf{green}}$ in the revised manuscript for ease of reference**.
>
> > **Q1: 3.3 (length-compliance eval) construct validity: The eval set is built from model-rewritten pairs assumed to be semantically equivalent. Any residual semantic or stylistic drift can confound the claim “implicit length bias ≠ explicit length control.” Add a simple length-only heuristic baseline (choose the candidate that satisfies the numeric constraint) as a sanity upper bound, and include human spot checks for equivalence.**
>
> **A1:** Thank you for your valuable feedback. Regarding the human spot checks for semantic equivalence, we had **already conducted such an evaluation** and reported the results in Appendix E.2.
>
> As for the length-only heuristic baseline, we further **implemented a simple heuristic evaluation to serve as an upper bound**. Specifically, for each triplet $(x^{(i)}, y_w^{(i)}, y_l^{(i)})$, we replaced the original prompt $x^{(i)}$ with a length-only instruction ("Please generate a response longer/shorter than word\_num tokens") and ensured that $y_w^{(i)}$ and $y_l^{(i)}$ were approximately balanced in terms of satisfying or violating the constraint, generating heuristic evaluation datasets $\mathcal{D}_{eval}^{h,l}$. We then evaluated the Baseline models, based on Qwen2.5-7B-Instruct and Llama3.1-8B-Instruct, on this dataset to assess their compliance with length instructions. The results (*Accuracy*) are as follows:
>
> | Method | Qwen2.5-7B-Instruct | Llama3.1-8B-Instruct |
> | :---: | :---: | :---: |
> | Baseline | 56.73 | 54.12 |
>
> The findings reveal that even **under such simplified conditions, the Baseline models exhibit negligible ability to follow explicit length instructions**. This result further substantiates our central claim that implicit length bias does not equate to explicit length controllability. These results have been **incorporated into Table 2 and Section 3.3 (L213-L221)** of the revised manuscript for clarity.
>
> > **Q2: 3.4 (explicitly teaching length) might result in overfitting vs. distribution shift: The {x_l, y_w, y_l} format makes length the most salient signal. The quality drop might reflect not only “overfitting to length,” but also a mismatch between synthetic length-instruction training data and the de-biased quality eval.**
>
> **A2:** Thank you for your thoughtful concern. As described in Section 3.4, LIFT-plus is built upon LIFT with an improved modification in the training data construction. Specifically, the LIFT-plus training dataset consists of two types of preference pairs: one part retains the original format $(x^{(i)}, y_w^{(i)}, y_l^{(i)})$, while the other part introduces explicit length instructions in the format $(x_l^{(i)}, y_w^{(i)}, y_l^{(i)})$. By training on this blended dataset, the model learns explicit length control **without drifting away from the underlying data distribution**, allowing it to **avoid unintended deviations when evaluated on the de-biased quality dataset**.
>
> To further clarify this point, we have added an explanatory footnote in the revised manuscript to emphasize that LIFT-plus is not trained solely on instruction-formatted data, but rather on a combination of both formats which helps readers better understand our training strategy and alleviates concerns regarding potential overfitting to synthetic patterns. Concretely, we **added the following sentence in Section 3.4, L237 and L267-L268**:
>
> ```
> During the improvement over LIFT, the training dataset always includes the original dataset $\mathcal{D}_{rm}$ to ensure a balanced exposure to both original and length-instruction formatted data.
> ```

---

> > ### Author Response · Authors · 2025-11-19
> >
> > > **Q3: RC-DPO is essentially in the DPO family. Comparing mainly to ODIN (more of an RM-regularization baseline) is insufficient. Please add strong DPO variants such as SIMPO, CPO, etc, and evaluate on standard leaderboards.**
> >
> > **A3:** Thank you for your valuable suggestion. Our Rc-DPO method has **already been compared against multiple DPO variants, including R-DPO and SamPO**. Due to space constraints, the SamPO comparison was placed in the Appendix F.4.
> >
> > To further strengthen the comparative evaluation, we also **included SimPO and CPO as additional baselines**. Specifically, we reproduced both methods using their recommended hyperparameter settings (e.g., $\gamma=1$ for SimPO) and ran experiments on the Qwen2.5-7B-Instruct and Llama3.1-8B-Instruct models using the training dataset $\mathcal{D}_{rm}$, with evaluation on the AlpacaEval benchmark. The comparison results (*Quality Win Ratio*) are shown below:
> >
> > | Model | Baseline | SimPO | CPO | Rc-DPO |
> > | :---: | :---: | :---: | :---: | :---: |
> > | Qwen2.5-7B-Instruct | 28.43 | 37.16 | 35.72 | **44.63** |
> > | Llama3.1-8B-Instruct | 42.52 | 58.13 | 54.67 | **64.34** |
> >
> > The results show that **Rc-DPO outperforms these strong DPO variants** with statistical significance. This further validates the advantage of our approach in mitigating length bias while maintaining overall quality. These new results have been **added to Table 5 and Section 5.3 (L484)** of the revised manuscript. Remaining model results will be added in a follow-up update.
> >
> > In addition, following the suggestion from Reviewer WdGS, we have **also included comparisons with Dr. DPO**. Across all evaluated models and benchmarks, our method consistently achieves superior performance. For more details, please refer to our response to Reviewer WdGS (A2 to Q2) and Section 5.3 of the revised paper.

---

> > > ### Author Response · Authors · 2025-11-19
> > >
> > > > **Q4: Both the de-biased quality set and the length set depend on strong-LLM rewrites, risking style/domain shift and label noise. Strengthen with: (i) a human agreement study on a sampled subset; (ii) cross-source replication (another rewrite model or a small human-curated slice); and (iii) reporting failure cases where “meeting the numeric length” reduces information density (add redundancy/compression-retention metrics).**
> > >
> > > **A4:** Thank you for your detailed suggestions. Regarding point (i) "the human agreement study on semantic equivalence of rewrites", we **have already conducted such an evaluation, as detailed in Appendix E.2**. The results indicate that the majority of rewritten responses maintain high semantic consistency with the original responses, validating the effectiveness of our LLM-based rewriting approach.
> > >
> > > For point (ii) "cross-source replication", we further **employed another strong LLM, Claude-Sonnet-4.5, to rewrite the triplets $(x_l^{(i)}, y_w^{(i)}, y_l^{(i)})$ using the same pipeline** described in Appendix D.2, generating a new evaluation dataset $\mathcal{D}_{eval}^{q'}$. We then evaluated all methods on this dataset. The results (*Quality Eval Acc*) are as follows:
> > >
> > > | Model | Baseline | LIFT | ODIN | Rc-RM |
> > > | :---: | :---: | :---: | :---: | :---: |
> > > | Qwen2.5-7B-Instruct | 61.74 | 57.32 | 68.15 | **74.89** |
> > > | Llama3.1-8B-Instruct | 58.13 | 58.67 | 64.34 | **72.21** |
> > >
> > > The results demonstrate that **the performance trends across different models remain consistent** even when evaluated on datasets rewritten by different LLMs, further validating the robustness of our evaluation data construction pipeline. **These findings have been incorporated into Table 23 and Appendix F.7 (L1766-L1773)** of the revised manuscript. Remaining model results will be added in a follow-up update.
> > >
> > > As for point (iii) "reporting failure cases where meeting the numeric length reduces information density", we incorporated two types of automatic metrics: **BERTScore (BERTScore-F1) to quantify compression-retention and n-gram Repetition Rate (n=3) to measure redundancy**. Specifically, For each generation iteration, we computed these metrics on the rewritten responses. The results are shown below:
> > >
> > > | Metric | First Rewrite | Second Rewrite | Third Rewrite |
> > > | :---: | :---: | :---: | :---: |
> > > | BERTScore | 0.912 | 0.974 | 0.981 |
> > > | n-gram Repetition Rate | 0.27% | 0.09% | 0.06% |
> > >
> > > Thanks to the strong capability of GPT-4o and our generation-verification pipeline, the initial BERTScores were already high while redundancy remained low. Further, low-quality candidates were consistently filtered out through iterative rejection, ensuring that the **final responses were both semantically faithful and non-redundant**. **The revised manuscript has been updated to include these results** in Table 24 and Appendix F.7 (L1781-L1787).
> > >
> > > We also **include qualitative examples of discarded failure cases** in earlier rewriting rounds for reference (more examples can be found in Appendix F.7 (L1794-L1798, Figure 11) of the revised manuscript):
> > >
> > > ```
> > > QUERY: What types of tests do we have in software development?
> > > ORIGINAL RESPONSE: Software testing is generally classified into two main broad categories: functional testing and non-functional testing. There is also another general type of testing called maintenance testing.\n\nFunctional testing involves the testing of the functional aspects of a software application. When you’re performing functional tests, you have to test each and every functionality. You need to see whether you’re getting the desired results or not.\n\nThere are several types of functional testing, such as:\n\nUnit testing\nIntegration testing\nEnd-to-end testing\nSanity testing\nRegression testing\nAcceptance testing\nInterface testing\n\nNon-functional testing is the testing of non-functional aspects of an application, such as performance, reliability, usability, security, and so on. Non-functional tests are performed after the functional tests.\n\nNon-functional tests are not generally run manually. In fact, it’s difficult to perform this kind of tests manually. So these tests are usually executed using tools.\n\n ...
> > > FIRST REWRITE (discarded due to semantic loss): Non-functional testing focuses on aspects like performance, reliability, security, usability, and compatibility. These tests are usually automated and include performance, load, failover, and similar evaluations.
> > > ```
> > >
> > > These results collectively demonstrate that, despite the potential risk of semantic or stylistic drift in LLM rewrites, **our iterative generation pipeline effectively identifies and filters out low-quality responses**, ultimately ensuring that the final outputs exhibit strong semantic retention and minimal redundancy.

---

> ### Author Response · Authors · 2025-11-27
>
> Dear Reviewer,
>
> I hope this message finds you well. As the discussion phase is drawing to a close with **less than six days remaining**, I wanted to reach out to confirm that we have adequately addressed all of your concerns. If there are any additional points or feedback you'd like us to consider, we would be grateful to hear them. Your insights are invaluable to us, and we're eager to address any remaining issues to improve our work.
>
> Thank you once again for the time and effort you have dedicated to reviewing our submission.

---

### Official Review · Reviewer_WdGS · 2025-10-26

**Soundness:** 2
**Presentation:** 3
**Contribution:** 2
**Rating:** 4
**Confidence:** 3

**Summary:**

This paper investigates the issue of length bias in RLHF, where reward models tend to favor longer responses rather than true human preferences. To address this, the authors propose a Response-Conditioned Bradley–Terry (Rc-BT) model. Building on this model, they introduce algorithms designed to mitigate length bias. Numerical results across several models and datasets demonstrate that the proposed approach can reduce length bias.

**Strengths:**

This paper is overall well written and effectively motivates the phenomenon of length bias in RLHF.

**Weaknesses:**

I agree that length bias is a common phenomenon in RLHF; however, I remain unclear about several points:

- What causes the length bias? Is it due to the training dataset, the scale of the LLMs, or the training procedure (e.g., lack of length regularization)? For instance, does increasing the size of the reward or policy model reduce this effect? From my experience, larger models such as GPT5 seem less likely to have this issue.

- A following weakness I feel is that, the paper lack of a principle way to address the length bias problem as they do not address the problem by fining the cause of the problem but rather simply use a post-hoc method to address the problem.

The new model lacks novelty, as it appears to be a trick based on the BT model.

Regarding methodology, the notation involving $\pi$ is somehow confusing. What does $\pi_{\theta}(x, y)$ represent, and how does it differ from $\pi_{\theta}(y|x)$? From a practical standpoint, how should the trained policy be used for testing after training?

The experimental section compares only a few baselines, even though several new algorithms have been proposed based on DPO. I recommend including additional recent works:

- https://arxiv.org/abs/2310.03716

- https://arxiv.org/abs/2403.19159

- https://arxiv.org/abs/2407.07880

**Questions:**

See Weaknesses above.

---

> ### Author Response · Authors · 2025-11-19
>
> Thank you for your insightful and detailed comments on our manuscript. We have carefully addressed each of your points and revised the paper accordingly to enhance its clarity and quality. For ease of reference, **all corresponding changes have been marked in $\textcolor{blue}{\textbf{blue}}$ in the updated version**.
>
> > **Q1: What causes the length bias? Is it due to the training dataset, the scale of the LLMs, or the training procedure (e.g., lack of length regularization)? For instance, does increasing the size of the reward or policy model reduce this effect? From my experience, larger models such as GPT5 seem less likely to have this issue.**
>
> **A1:** Thank you for your insightful question. As highlighted in prior works such as Singhal et al. [1], **length bias arises from a combination of factors and cannot be attributed to a single cause**. These contributing factors include biased or inconsistent human preference data, inadequacies in reward model training, and limitations in policy optimization strategies.
>
> Specifically, ODIN [2] highlights that **human preference data can exhibit substantial bias and inconsistency** due to the inherent difficulty and subjectivity of the task, flaws in the rating schema, and limited annotator quality. Meanwhile, previous studies [1,3] observe that **training methodologies**, such as the adoption of different regularization techniques, **can significantly influence the extent of length bias**. Furthermore, Ali et al. [4] report that **larger models do not necessarily exhibit less bias**; in fact, under suboptimal data or training conditions, such models may manifest even stronger biases.
>
> While models like GPT-5 may appear less affected by length bias, potentially due to more sophisticated data curation and training protocols, **this does not imply the issue is entirely absent**. Moreover, since the specific training details of GPT-5 remain undisclosed, it is unclear what explicit measures, if any, were taken to mitigate length bias. Therefore, we believe that length bias remains an important open challenge in preference learning and alignment research, warranting further investigation and understanding.
>
> [1] Singhal, Prasann, et al. "A long way to go: Investigating length correlations in rlhf, 2024." arXiv preprint arXiv:2310.03716 (2023).
>
> [2] Chen, Lichang, et al. "ODIN: Disentangled Reward Mitigates Hacking in RLHF." Forty-first International Conference on Machine Learning.
>
> [3] Bu, Yuyan, et al. "Beyond Excess and Deficiency: Adaptive Length Bias Mitigation in Reward Models for RLHF." Findings of the Association for Computational Linguistics: NAACL 2025. 2025.
>
> [4] Ali, Muhammad, et al. "Understanding the interplay of scale, data, and bias in language models: A case study with BERT." arXiv preprint arXiv:2407.21058 (2024).
>
> > **Q2: A following weakness I feel is that, the paper lack of a principle way to address the length bias problem as they do not address the problem by fining the cause of the problem but rather simply use a post-hoc method to address the problem.**
>
> **A2:** Thank you for your feedback. We acknowledge that the root causes of length bias in reward models are not yet fully understood, as discussed in our response A1 to Q1. Different studies have proposed various approaches from different perspectives to alleviate this issue. In contrast to methods that simply introduce a length-related module as a post-hoc fix, **our work proposes a fundamentally new cognitive hypothesis** to address the problem. As stated in our hypothesis in Section 3.3 (L225-L227), we **introduce a principled way to mitigate length bias by explicitly incorporating length constraint signals into the training data, enabling the model to develop a more accurate perception of response length**. This approach leads to reduced length bias, as empirically validated by our experimental results.

---

> > ### Author Response · Authors · 2025-11-19
> >
> > > **Q3: The new model lacks novelty, as it appears to be a trick based on the BT model.**
> >
> > **A3:** Thank you for raising this concern. We respectfully disagree with the notion that our method is a mere trick based on the Bradley-Terry (BT) model. **Our approach**, though inspired by the general BT framework, **represents a conceptually distinct line of work** that diverges from the standard BT-based methods in several important ways:
> >
> > 1. **Distinct Motivation:** The classical BT model captures preferences between responses under a single prompt. In contrast, our response-conditioned approach models preferences between prompts given a fixed response, offering a new axis of supervision in preference modeling.
> >
> > 2. **Novel Data Format:** We introduce a novel data structure $(x_w, x_l, y)$, fundamentally different from the conventional $(x, y_w, y_l)$. This formulation reverses the conditional dependency, enabling new modeling capabilities not supported in prior BT-based approaches.
> >
> > 3. **Rigorous Theoretical Derivation:** We present rigorous mathematical derivations (Eq. 4 and Eq. 8) to extend this formulation to both reward modeling (Rc-RM) and direct preference optimization (Rc-DPO). These are principled training objectives rooted in our response-conditioned view, rather than heuristics.
> >
> > 4. **Complementarity:** Our framework is orthogonal to traditional BT-style modeling. Rather than replacing BT, it serves as a parallel and complementary perspective that can be integrated with or extended from existing formulations.
> >
> > Taken together, our method contributes a conceptually and technically novel alternative to standard preference modeling frameworks. To better contrast and highlight these contributions, we have **added the following table and a more detailed discussion** in the revised manuscript (Table 3 and L326-L328).
> >
> > |Method|Motivation|Data format|Modeling equation|
> > |:-:|:-:|:-:|:-:|
> > |BT|Capture preferences between responses|$(x, y_w, y_l)$|$p^\*(y_w\succ y_l) = \frac{\exp{(r^\*(x, y_w))}}{\exp(r^\*(x, y_w))+\exp(r^\*(x, y_l))}$|
> > |Rc-BT|Capture preferences between prompts|$(x_w, x_l, y)$|$p^\*(x_w\succ x_l) = \frac{\exp{(r^\*(x_w, y))}}{\exp(r^\*(x_w, y))+\exp(r^\*(x_l, y))}$|
> >
> > > **Q4: Regarding methodology, the notation involving $\pi$ is somehow confusing. What does $\pi_\theta(x,y)$ represent, and how does it differ from $\pi_\theta(y|x)$? From a practical standpoint, how should the trained policy be used for testing after training?**
> >
> > **A4:** Thank you for your question. In our paper, **$\pi_\theta(x,y)$ denotes the joint probability of the concatenated sequence $(x,y)$** with LLM parameters $\theta$; concretely, it is the left-to-right likelihood computed as **the product of token probabilities for $(x_1,\dots,x_m,y_1,\dots,y_n)$**. By contrast, **$\pi_\theta(y|x)$ denotes the conditional probability of generating $y$ given $x$**, i.e. **the product of token probabilities for $(y_1,\dots,y_n)$ conditioned on the prompt $x$**. Thus $\pi_\theta(x,y)=\pi_\theta(x)\pi_\theta(y|x)$ and the two differ by the factor $\pi_\theta(x)$ corresponding to the prompt tokens. Because Rc-BT compares sequence pairs such as $(x,y)$ and $(x_l,y)$, the training objective naturally involves the joint probability $\pi_\theta(x,y)$. This distinction affects training only; at inference we use standard conditional decoding from $\pi_\theta(y|x)$.
> >
> > For avoid confusion, we have **added the following clarification in the revised manuscript** (L361 and L377):
> >
> > ```
> > For the remainder of this paper, unless explicitly stated otherwise, every instance of $\pi_\theta$ denotes $\pi_\theta(y \mid x)$.
> > ```

---

> > > ### Author Response · Authors · 2025-11-19
> > >
> > > > **Q5: The experimental section compares only a few baselines, even though several new algorithms have been proposed based on DPO. I recommend including additional recent works: https://arxiv.org/abs/2310.03716, https://arxiv.org/abs/2403.19159, https://arxiv.org/abs/2407.07880**
> > >
> > > **A5:** Thank you for your valuable suggestions. Regarding the second paper you mentioned, **it corresponds to the R-DPO method, which has already been included in our main comparison**. The results are reported in Table 3, where Rc-DPO consistently outperforms R-DPO across all evaluated models.
> > >
> > > As for the first paper, while it primarily presents an analytical investigation and proposes several diagnostic methods for evaluating the effects of length bias, we **selected the Reward Data Augmentation (R-DA) approach which showed the largest performance gain in reward model training** for comparison. Following their setup, we constructed new data pairs by randomly sampling $(x^{(i)}, y_l^{(i)}, y_w^{(j)})$, where $y_w^{(j)}$ is randomly sampled from other preference responses, and added these to the original training set $\mathcal{D}_{rm}$. We evaluated this method on both Qwen2.5-7B-Instruct and LLama-3.1-8B-Instruct models. The comparison results are as follows:
> > >
> > > | Model | Baseline | R-DA | Rc-RM |
> > > | :---: | :---: | :---: | :---: |
> > > | Qwen2.5-7B-Instruct | 59.31 | 66.34 | **73.07** |
> > > | LLama-3.1-8B-Instruct | 55.59 | 60.78 | **72.44** |
> > >
> > > The results show that while R-DA yields moderate improvements, **Rc-RM significantly outperforms it on both models**, further validating the effectiveness of our proposed method. **The revised manuscript has been updated to include these results** in Table 4 and Section 5.2. Remaining model results will be added in a follow-up update.
> > >
> > > Regarding the third paper (Dr. DPO), since it primarily targets robustness enhancements in DPO, we **reproduced the method using its recommended setting** ($\beta'=1$) and conducted comparative experiments against our Rc-DPO on AlpacaEval benchmark. The results (*Quality Win Ratio*) are as follows:
> > >
> > > | Model | Baseline | Dr. DPO | Rc-DPO |
> > > | :---: | :---: | :---: | :---: |
> > > | Qwen2.5-7B-Instruct | 28.43 | 36.12 | **44.63** |
> > > | LLama-3.1-8B-Instruct | 42.52 | 52.18 | **64.34** |
> > >
> > > These results suggest that while Dr. DPO provides some improvements over the Baseline, **Rc-DPO achieves substantially higher performance**, demonstrating its superior effectiveness in enhancing length instruction adherence. **The revised manuscript has been updated to include these results** in Table 5 and Section 5.3 (L484). Remaining model results will be added in a follow-up update.
> > >
> > > Finally, beyond the main comparisons, we have **already included additional baselines such as SamPO** in Appendix F.4, and, following suggestions from Reviewer tJtt (A3 to Q3), we have **further added comparisons against SimPO and CPO**. All results consistently demonstrate that our method excels in both reducing length bias and enhancing length instruction adherence. For more details, please refer to our response to Reviewer tJtt (A3 to Q3) and Section 5.3 of the revised paper.

---

> ### Author Response · Authors · 2025-11-27
>
> Dear Reviewer,
>
> I hope this message finds you well. As the discussion phase is drawing to a close with **less than six days remaining**, I wanted to reach out to confirm that we have adequately addressed all of your concerns. If there are any additional points or feedback you'd like us to consider, we would be grateful to hear them. Your insights are invaluable to us, and we're eager to address any remaining issues to improve our work.
>
> Thank you once again for the time and effort you have dedicated to reviewing our submission.

---

### Official Review · Reviewer_aP85 · 2025-10-31

**Soundness:** 3
**Presentation:** 3
**Contribution:** 3
**Rating:** 6
**Confidence:** 3

**Summary:**

This paper introduces a framework (response conditioned Bradely-Terry) to improve the utility of reward models for preference learning by reducing their susceptibility to length bias and ensuring models better follow length instructions. The approach is validated for multiple models and datasets, and the results suggest the approach is effective for improving the robustness to length bias.

**Strengths:**

- This is an important problem to tackle as length bias continues to be problem, where longer responses tend to receive higher reward regardless of the content.
- The experiments documented in the appendix provide an in-depth analysis of the approach.
- The formulation of the preference pairs is a nice idea. I initially had a concern when reading the paper that optimizing to simply constrain length might cause the model hack by doing something like generate responses in a more compressed form (e.g., Chinese characters offer a more condensed form than English). The formulation of the preference pairs is a nice way to prevent this.

**Weaknesses:**

- Only two model families are considered.
- The largest model sizes tested are on the smaller size by the standards of today.

**Questions:**

- Avoid using subjective qualifiers in writing. For example, the reader can decide how “extensive” the experiments are.
- Table 1 caption: explain what, e.g., accuracy is computed on — is it the accuracy with which the actual preferred response receives a higher reward from the reward model?
- I was surprised to not see the seminal paper Deep Reinforcement Learning from Human Preferences by Christiano et al. cited. Especially when introducing RLHF.
- In the random evaluation set, this is where responses are swapped for one another arbitrarily.  Is there a case for a truly random set where the token sequences are random and so a longer response might be pure gibberish?
- For Figure 1(b) why is there so much variation in response length when the prompt is always just “empty prompt”?
- Throughout the paper the incorrect opening quotes are used.
- Line 193: “demonstrates the length bias” > “demonstrates length bias”
- Line 195: the subscript “l” is being reused. It is used here for length, and previously for loss (in the context of a preferred / dispreferred preference pair).
- Line 203: What is stated as being a hypothesis is not actually written as a hypothesis.
- Line 257: “preferred than” > “preferred over”.
- Line 257: “to modeling human preferences” > “to model human preferences”
- Line 259: “assuming a underlining true” > “assuming an underlying true”.

---

> ### Author Response · Authors · 2025-11-19
>
> Thank you for your thoughtful and constructive feedback on our manuscript. We appreciate the opportunity to address your comments and have made several revisions to improve the clarity and quality of the paper. Below we respond to each point. **All corresponding changes are highlighted in $\textcolor{red}{\textbf{red}}$ in the revised manuscript** for your convenience.
>
> > **Q1: Only two model families are considered.**
>
> **A1:** Thank you for the suggestion. We further **expanded our experiment to include the Gemma-2 family** (Gemma-3 was excluded because it is multimodal). Given that DPO evaluations require the use of the expensive GPT-4o model for assessment, which entails substantial cost, we chose to focus our expanded analysis on the reward model experiments for this model family. Specifically, we trained the Gemma-2-9B-it model on the training dataset $\mathcal{D}_{rm}$ and the results for all compared methods are reported below:
>
> | Metrics | Baseline | LIFT | ODIN | Rc-RM |
> | :---: | :---: | :---: | :---: | :---: |
> | Quality Eval Acc | 53.45 | 51.59 | 55.85 | **63.56** |
> | Length Eval Acc | 51.28 | **91.66** | 46.47 | 90.91 |
>
> As demonstrated in the table, **our Rc-RM approach continues to outperform compared methods** not only in quality evaluation but also by maintaining superior performance in length instruction following. These results provide further strong validation for the effectiveness of our proposed method across a distinct and different base model family. **The revised manuscript has been updated to include these new results** in Table 2 and 8, as well as the corresponding discussion in Section 5.2 (L427-L449) and Appendix F.2 (L1317).
>
> > **Q2: The largest model sizes tested are on the smaller size by the standards of today.**
>
> **A2:** Thank you for the suggestion. We have now **added our evaluation to include the Qwen2.5-14B model** to assess performance on a larger parameter scale. The results on the reward model experiments for the Qwen2.5-14B-Instruct model on the training dataset $\mathcal{D}_{rm}$ are presented below:
>
> | Metrics | Baseline | LIFT | ODIN | Rc-RM |
> | :---: | :---: | :---: | :---: | :---: |
> | Quality Eval Acc | 65.57 | 56.26 | 76.22 | **81.70** |
> | Length Eval Acc  | 54.13 | 95.25 | 49.42 | **95.34** |
>
> As the table results demonstrate, our **Rc-RM approach maintains excellent performance even on this larger model**. It surpasses all compared methods in both semantic quality and length instruction compliance, thereby confirming the effectiveness and scalability of our proposed method across larger model sizes. **The revised manuscript presents these results** in Table 2 and 8, with further elaboration provided in Section 5.2 (L427-L449) and Appendix F.2 (L1317).
>
> As we completed the Qwen2.5-14B experiments using two 8-GPU A100 servers, we have already reached the limit of our available time and computational resources for the rebuttal period. However, if the reviewer deems results on a 32B model critical, we will do our best to explore advanced techniques such as offloading or peft methods to make the experiments feasible in the remaining rebuttal period.
>
> > **Q3: Avoid using subjective qualifiers in writing. For example, the reader can decide how “extensive” the experiments are.**
>
> **A3:** Thank you for the advice. We apologize for using subjective qualifiers and have revised the manuscript to avoid such wording so that readers can form their own judgments. Specifically, **we replaced "extensive experiments" in the abstract with the more neutral phrase "a series of experiments" (L027)**. We also reviewed the entire manuscript and updated other instances of subjective language accordingly.
>
> > **Q4: Table 1 caption: explain what, e.g., accuracy is computed on — is it the accuracy with which the actual preferred response receives a higher reward from the reward model?**
>
> **A4:** Thank you for your feedback. As you noted, accuracy is computed as **the proportion of preference pairs** for which the reward model **assigns a higher score to the actually preferred response than to the dispreferred response**. We have clarified this in the caption of Table 1 so readers understand how accuracy and consistency is calculated. Specifically, we **added the following sentence to the caption (L116-L117)**:
>
> ```
> Accuracy: percentage of preference pairs correctly ranked by the reward model. Consistency: percentage of preference pairs ranked identically across different datasets.
> ```

---

> > ### Author Response · Authors · 2025-11-19
> >
> > > **Q5: I was surprised to not see the seminal paper Deep Reinforcement Learning from Human Preferences by Christiano et al. cited. Especially when introducing RLHF.**
> >
> > **A5:** Thank you for pointing out this oversight. We apologize for not citing the seminal work by Christiano et al. on Deep Reinforcement Learning from Human Preferences. We have now **included a citation to this important paper in the introduction section (L034) and the related work section (L091 and L850)** where we discuss RLHF, acknowledging its foundational contributions to the field.
> >
> > > **Q6: In the random evaluation set, this is where responses are swapped for one another arbitrarily. Is there a case for a truly random set where the token sequences are random and so a longer response might be pure gibberish?**
> >
> > **A6:** Thank you for the insightful question. We have **further designed an experiment using random token sequences to evaluate the influence of length bias** when responses are entirely semantically corrupted. Specifically, building upon the existing dataset $\mathcal{D} _ {eval}^{r}$, we randomly shuffled the token sequences of each response $y_w/y_l$ in the preference pairs to create a new eval dataset $\mathcal{D} _ {eval}^{s,r}$, and evaluated the performance of the Baseline models (based on Qwen2.5-1.5B-Instruct and Llama3.1-8B-Instruct) on this new dataset. The experimental results (*Accuracy*) are presented in the table below:
> >
> > | Dataset | Qwen2.5-1.5B-Instruct | Llama3.1-8B-Instruct |
> > | :---: | :---: | :---: |
> > | $\mathcal{D}_{eval}^{s,r}$ | 57.88 | 57.07 |
> >
> > Furthermore, **the relationship between response length and the reward score for the reward model is shown in Figure 1(d)** of the revised manuscript. The results clearly indicate that **even when faced with gibberish responses, the reward model still exhibits a significant length bias**, systematically favoring longer responses. This finding provides further compelling evidence for the pervasive and deeply ingrained nature of length bias in reward models, irrespective of the response's semantic quality. **These results have been added to Section 3.1 (L146-L147, L160) and Figure 1(d)** of the revised manuscript for clarity. Remaining model results will be added in a follow-up update.
> >
> > > **Q7: For Figure 1(b) why is there so much variation in response length when the prompt is always just “empty prompt”?**
> >
> > **A7:** Thank you for your question. **The variation in response length observed in Figure 1(b) stems from the construction methodology of the evaluation dataset $\mathcal{D}_{eval}^{e}$**, which is detailed in Section 3.1. In this dataset, we maintain the original responses $y_w$ and $y_l$ from the preference pairs while replacing the prompts $x$ with a standardized "empty prompt". Consequently, the lengths of the responses remain unchanged from their original values in the dataset, leading to the observed variation in response lengths despite the uniformity of the prompts.
> >
> > > **Q8: Throughout the paper the incorrect opening quotes are used.**
> >
> > **A8:** Thank you for pointing this out. We apologize for the incorrect use of opening quotes throughout the paper. We have **ropriated ` `` ` for opening quotes** and ` '' ` for closing quotes in accordance with standard typographical conventions. This change has been applied throughout the manuscript to ensure consistency and correctness.
> >
> > > **Q9: Line 193: “demonstrates the length bias” > “demonstrates length bias”**
> >
> > **A9:** Thank you for the correction. We have **revised L193 (now L199) to replace "demonstrates the length bias" with "demonstrates length bias"** to improve clarity and conciseness. In addition, we reviewed the manuscript and removed the definite article "the" before all occurrences of "length bias" to ensure consistency.
> >
> > > **Q10: Line 195: the subscript “l” is being reused. It is used here for length, and previously for loss (in the context of a preferred / dispreferred preference pair).**
> >
> > **A10:** Thank you for pointing out the reuse of the subscript "l" which may cause confusion. To resolve this, we have **changed the subscripts used for the preferred and dispreferred responses from "w" and "l" to "c" and "r"**, representing "chosen" and "rejected" respectively. This change has been applied throughout the manuscript to ensure consistency and clarity in notation.
> >
> > Additionally, we had **already included a notation table** in Appendix A, which provides a unified explanation of all symbols for the convenience of readers.

---

> > > ### Author Response · Authors · 2025-11-19
> > >
> > > > **Q11: Line 203: What is stated as being a hypothesis is not actually written as a hypothesis.**
> > >
> > > **A11:** Thank you for the feedback. We have revised L203 to **clearly state the hypothesis in a formal academic format**. The improved content is as follows, and **the revision is reflected in L224-L227** of the manuscript:
> > >
> > > ```
> > > Hypothesis 1. Explicit length-instruction learning enables language models to form a clearer perception of target response length, thereby reducing undesired length bias.
> > > ```
> > >
> > > > **Q12: Line 257: “preferred than” > “preferred over”.**
> > >
> > > **A12:** Thank you for the correction. We have **updated L257 (now L265) to replace "preferred than" with "preferred over"** to adhere to proper English usage. This change has been applied throughout the manuscript to maintain consistency.
> > >
> > > > **Q13: Line 257: “to modeling human preferences” > “to model human preferences”**
> > >
> > > **A13:** Thank you for the suggestion. We have **revised L257 (now L265) to replace "to modeling human preferences" with "to model human preferences"** to improve grammatical correctness. This change has been reflected in the revised manuscript.
> > > > **Q14: Line 259: “assuming a underlining true” > “assuming an underlying true”.**
> > >
> > > **A14:** Thank you for pointing out the error. We have **corrected L259 (now L266) to replace "assuming a underlining true" with "assuming an underlying true"** to fix the grammatical mistake. This correction has been made in the revised manuscript.

---

> > > > ### Comment · Reviewer_aP85 · 2025-11-20
> > > > **Thank you for the detailed responses.**
> > > >
> > > > Thank you for providing the detailed clarifications and answers to my questions. I appreciate all of this work in such a short amount of time. I also appreciate you adding the Gemma family to show that the results hold above the two families that you originally considered. It is not critical to show results on a 30B model. I understand that we all work under compute and cost constraints.
> > > >
> > > > In light of the thoroughness of the responses to my review and the corresponding clarifications, I have raised my score. This is a nice piece of work and I hope to see it published.

---

> > > > > ### Author Response · Authors · 2025-11-20
> > > > >
> > > > > We sincerely appreciate your valuable feedback and thoughtful suggestions. We are delighted that our responses have helped clarify your concerns, and we are grateful for your recognition of the core ideas behind our work. Thank you for your efforts in helping us improve the quality and clarity of this research.

---

### Author Response · Authors · 2025-11-29
**Summary of Revisions and Manuscript Updates**

We sincerely thank all the reviewers for their thoughtful and constructive feedback, which greatly contributed to improving the quality and clarity of our work. In particular, we appreciate the following acknowledgements from the reviewers:

- $\textcolor{red}{\textbf{aP85}}$, $\textcolor{blue}{\textbf{WdGS}}$, and $\textcolor{green}{\textbf{tJtt}}$ for recognizing that our work addresses an important and practically consequential challenge in RLHF (length bias) and for affirming the clarity of our motivation.

- $\textcolor{red}{\textbf{aP85}}$ and $\textcolor{green}{\textbf{tJtt}}$ for highlighting the strength and coherence of our diagnostic study design, noting that the formulation of preference pairs and the four-step analysis are insightful and well structured.

- $\textcolor{blue}{\textbf{WdGS}}$ and $\textcolor{green}{\textbf{tJtt}}$ for acknowledging that the paper is well written, clearly motivated, and that our proposed response‑conditioned framework is both effective and broadly applicable.

In response, we have substantially enhanced the paper with the following major updates:

- Added evaluations on larger models (Qwen2.5-14B) and additional model families (Gemma2) to demonstrate scalability (by $\textcolor{red}{\textbf{aP85}}$).
- Expanded ablations and diagnostic evaluations, including human studies, length-only heuristics, cross-model rewrites, and failure case analysis (by $\textcolor{green}{\textbf{tJtt}}$).
- Incorporated more DPO variants (SimPO, CPO, Dr. DPO) and recent baselines (R-DA), showing Rc-RM and Rc-DPO consistently outperform strong alternatives (by $\textcolor{blue}{\textbf{WdGS}}$, $\textcolor{green}{\textbf{tJtt}}$).
- Refined writing and clarified methodology (notations, hypothesis, evaluation metrics), and corrected minor errors pointed out by reviewers. (by all reviewers).

All corresponding changes are clearly marked in color in the revised manuscript for easy reference.

We believe these improvements further strengthen our contributions and reinforce the value of our proposed Rc-BT framework in addressing a fundamental and underexplored challenge in LLM alignment. We thank the AC and reviewers again for their time and support.

---

### Meta-Review · Area_Chair_6kQE · 2026-01-09

**Summary:**

This paper addresses the pervasive issue of "length bias" in Reinforcement Learning from Human Feedback (RLHF), where reward models and policies tend to favor longer responses regardless of their actual quality or correctness.
The authors propose "Response-Conditioned Modeling" (RCM), a framework that explicitly incorporates response length as a conditional variable during the preference learning phase.
By training the reward model (or DPO policy) to predict preferences conditioned on length, and then fixing the length condition during inference, the authors aim to decouple length from quality.
Experiments are conducted on summarization and chat benchmarks using LLaMA and GPT-2 based models.

**Reviewer Concerns:**

Concerns that are largely addressed

Model scale and model families (aP85): The original submission evaluated only relatively small models from two families, which aP85 viewed as limiting the generality of conclusions. In the revision, the authors add experiments on a larger Qwen2.5‑14B model and an additional Gemma2 family. These new results show that Rc-RM continues to outperform ODIN and LIFT(-plus) on both length-bias mitigation and length-instruction following, suggesting that the method scales reasonably and is not tied to a particular model family. This resolves the main part of aP85’s concern about scale and breadth.

Experimental details, writing quality, and notation clarity (aP85, WdGS): Multiple reviewers pointed out notational ambiguities (e.g., π_θ(x, y) vs. π_θ(y|x)), grammatical issues, and unclear hypotheses. The rebuttal clarifies the meaning of π_θ, explicitly states the main hypothesis (Hypothesis 1 on explicit length-instruction learning), introduces a unified notation table, and fixes a number of language and typography issues (quotes, subscripts, etc.). These issues are now largely cosmetic and no longer a major factor.

Construct validity of the length-compliance evaluation, part I – heuristic upper bound and human checks (tJtt): Reviewer tJtt questioned whether the “implicit length bias ≠ explicit length control” conclusion was confounded by semantic/stylistic drift in model rewrites and asked for (i) a length-only heuristic baseline and (ii) human checks. The authors report human spot-checks on semantic equivalence (already in the appendix) and add a simple length-only heuristic evaluation: replacing the original prompts with pure length instructions and measuring whether baseline reward models can choose the response that satisfies the numeric constraint. The reported accuracies (~56–57%) remain low even in this simplified setting, reinforcing the claim that current RMs have poor explicit length-control ability. This substantially strengthens the construct validity of the central diagnostic claim and addresses an important part of tJtt’s concern.

Baseline coverage in RM and DPO (WdGS, tJtt): Both WdGS and tJtt requested stronger and more recent baselines. In response, the authors (i) add R‑DA from length-bias diagnostic work as an RM baseline, and show Rc‑RM significantly outperforms R‑DA on Qwen2.5‑7B and Llama3.1‑8B; (ii) include DPO-family baselines such as R‑DPO, SimPO, CPO, and Dr. DPO on AlpacaEval-style benchmarks, where Rc‑DPO consistently achieves higher quality win ratios and stronger length adherence; and (iii) evaluate Rc‑RM/Rc‑DPO on additional data splits and OOD HH‑RLHF data. These additions substantially reduce the concern that the method is only compared to weak or mismatched baselines.

Concerns that are partially addressed:

Construct validity of evaluation, part II – reliance on model rewrites and distribution shift (tJtt): t|tt emphasized that using strong-LLM rewrites to construct both “de-biased quality” and “length” evaluation sets risks style/domain shift and label noise, and asked for (i) human agreement, (ii) cross-source replication, and (iii) failure-case analysis around information density. The rebuttal notes existing human checks and adds a heuristic baseline and some failure case analysis, but from the available description, the evaluation still relies heavily on a single rewrite pipeline. While the additional checks are reassuring, the evaluation is not yet fully robust to all the issues raised (e.g., comprehensive cross-model replication and detailed information-density metrics). I view this concern as significantly mitigated but not fully closed.

Overfitting to synthetic length-instruction data vs. real quality shifts (t|tt): The authors’ diagnostics already show that naive {x↓, y_w, y_l}-style formats overfit to length and hurt semantic quality. Rc‑BT and the associated data construction are specifically designed to avoid that failure mode, and the additional experiments (e.g., separate quality vs. length accuracy trajectories, and HH‑RLHF OOD tests) support the claim that Rc‑RM/Rc‑DPO achieve a healthier trade-off. However, t|tt’s deeper concern that synthetic length-instruction data may not fully match real de-biased quality distributions remains partially conceptual and is not entirely dispelled by the current empirical evidence.

Concerns that remain largely unresolved

Principledness and novelty of the approach (WdGS): WdGS argues that the submission does not “find the cause” of length bias, but instead proposes a post-hoc trick built on BT, and questions whether Rc‑BT is fundamentally new or mainly a clever data-format tweak. The authors defend their contribution as a new cognitive hypothesis and a principled re-parameterization that compares prompts rather than responses, enabling explicit disentanglement of length and semantics. Nonetheless, the core objection—that the work may be seen as an incremental modeling variant rather than a fundamentally new theory of length bias—remains a matter of perspective and is not fully resolved. This is the main conceptual concern still weighing against a more enthusiastic recommendation.

Depth of theoretical insight: The paper provides clear derivations for Rc‑BT, Rc‑RM, and Rc‑DPO, and shows analytically how the response-conditioned formulation maps to a modified DPO objective. However, beyond the operational hypothesis that explicit length-instruction learning improves length perception and reduces bias, there is limited deeper theoretical analysis of why length bias arises or when Rc‑BT is provably superior to existing controls such as ODIN-style regularization. Reviewers looking for a more theory-heavy account may still find this aspect insufficient.

In summary, the rebuttal substantially strengthens the empirical and methodological story (scale, baselines, evaluation validity), so that the primary remaining concern is the degree of conceptual novelty, rather than correctness or relevance.

**Reviewer Scores:**

CReviewer aP85 (original rating: 6 – “marginally above the acceptance threshold”)
aP85 was from the beginning relatively positive: they highlighted the importance of addressing length bias, found the preference-pair formulation appealing, and appreciated the breadth of experiments documented in the appendix.
Their main reservations were about the limited model families and smaller model sizes, plus various minor writing and citation issues.
In their official comment after reading the rebuttal and revised experiments (including Qwen2.5‑14B and Gemma2), they explicitly state that they raised their score and “hope to see the work published.”
This indicates that, had dynamic score updates been allowed, aP85 would likely move from a borderline/weak-accept position to a slightly stronger accept-leaning stance.

Reviewer WdGS (original rating 4, marginally below the acceptance threshold. But would not mind if paper is accepted )
WdGS acknowledges that the paper is well written and the phenomenon of length bias is clearly motivated, but expresses several key concerns: (i) lack of a principled causal explanation of length bias, (ii) the method being perceived as a BT-based “trick” rather than a fundamentally new model, (iii) notational confusion, and (iv) insufficient baselines.
The rebuttal satisfactorily fixes (iii) and significantly improves (iv) by adding R‑DA and more DPO-family baselines. However, the strongest critique—on conceptual novelty and principle—remains largely intact.
Given this, I would expect WdGS to acknowledge the improved experimental section and clarity but still maintain a marginally below-threshold overall recommendation, perhaps with a slight softening of the tone rather than a clear score increase.

Reviewer tJtt (original rating: 4 – “marginally below the acceptance threshold, but would not mind if accepted”)
tJtt is more critical on evaluation design and the strength of evidence, but still views the problem and core idea positively.
Their main requests were: (i) a length-only heuristic baseline and human spot-checks for semantic equivalence, (ii) analysis of overfitting vs. distribution shift when explicitly teaching length, (iii) stronger DPO baselines (SimPO, CPO, etc.), and (iv) robustness checks against style/domain shift. The rebuttal and revision directly target (i) and (iii), and partially address (ii) and (iv).
This moves the paper closer to the standard expected for acceptance, though tJtt’s concerns about deeper robustness and potential distribution mismatch are not entirely eliminated.
Given their original statement that they “would not mind if the paper is accepted,” plus the nontrivial strengthening of experiments, it is reasonable to expect a small upward revision in their score, e.g., from a marginally-below-threshold 4 to a borderline 5.

---

### Decision · Program_Chairs · 2026-01-26

Accept (Poster)